

# Variability and evolution of mid-latitude stratospheric aerosol budget from 22 years of ground-based lidar and satellite observations

Sergey M. Khaykin[1], Sophie Godin-Beekmann[1], Philippe Keckhut[1], Alain Hauchecorne[1], Julien Jumelet[1], Jean-Paul Vernier[2,3], Adam Bourassa[4], Doug A. Degenstein[4], Landon A. Rieger[4], Christine Bingen[5], Filip Vanhellemont[5], Charles Robert[5], Matthew DeLand[6], Pawan K. Bhartia[7]

[1] LATMOS/IPSL, UVSQ Université Paris-Saclay, UPMC Univ. Paris 06, CNRS, Guyancourt, France.
[2] Science Systems and Applications, Inc., Hampton, Virginia, US.
[3] NASA Langley Research Center, Hampton, Virginia, US.
[4] Institute of Space and Atmospheric Studies, University of Saskatchewan, Saskatoon, Saskatchewan, Canada.
[5] Royal Belgian Institute for Space Aeronomy, Brussels, Belgium
[6] Science Systems and Applications, Inc., Lanham, Maryland, USA,
[7] NASA Goddard Space Flight Center, Greenbelt, Maryland, USA

*Correspondence to*: Sergey M. Khaykin (sergey.khaykin@latmos.ipsl.fr)

## Abstract

The article presents new high-quality continuous stratospheric aerosol observations spanning 1994-2015 at the French Observatoire de Haute-Provence (OHP, 44 °N, 6 °E) obtained by two independent regularly-maintained lidar systems. Lidar series are compared with global-coverage observations by Stratospheric Aerosol and Gas Experiment (SAGE II), Global Ozone Monitoring by Occultation of Stars (GOMOS), Optical Spectrograph and InfraRed Imaging System (OSIRIS), Cloud-Aerosol Lidar with Orthogonal Polarization (CALIOP) and Ozone Mapping Profiling Suite (OMPS) satellite instruments, altogether covering the time span of OHP lidar measurements.

Local OHP and zonal-mean satellite series of stratospheric aerosol optical depth are in excellent agreement, allowing for accurate characterization of stratospheric aerosol evolution and variability at Northern mid-latitudes during the post-Pinatubo era. The combination of local and global observations is used for careful separation between volcanically-perturbed and quiescent periods. While the volcanic signatures dominate the stratospheric aerosol record, the background aerosol abundance is found to be modulated remotely by poleward transport of convectively-cleansed air from the deep tropics and aerosol-laden air from the Asian monsoon region. The annual cycle of background aerosol at mid-latitudes, featuring a minimum during late spring and a maximum during late summer, correlates with that of water vapour from Microwave Limb Sounder (MLS).

Observations covering two volcanically-quiescent periods over the last two decades provide indication of a growth in the non-volcanic component of stratospheric aerosol. A statistically-significant factor of two increase of non-volcanic aerosol since 1998, seasonally restricted to late-summer and fall, is associated with the influence of the Asian monsoon and growing pollution therein.



## 1 Introduction

The role of stratospheric aerosol burden in climate variability and ozone chemistry is well recognized. Long-term observations of stratospheric aerosol are essential for interpretation of global atmospheric temperature and ozone layer variability (SPARC, 2006; Solomon et al., 2011). Regular vertically-resolved observations of stratospheric aerosol began in 1970s, 10 years after the pioneering in situ measurements by Junge et al. (1961) and remote detection by Fiocco and Grams
(1964). Global information on stratospheric aerosol is available since the late 1970s from various satellite missions, reviewed by SPARC (2006) and Kremser et al. (2016).

Volcanic eruptions with Volcanic Explosivity Index (VEI) $\geq 4$ injecting sulphur into the stratosphere are a major source of stratospheric aerosol. In the absence of strong eruptions, the permanent stratospheric aerosol layer (also termed background aerosol) is commonly attributed to
sulphuric gas precursors such as OCS and $SO_2$ emitted at the surface and lofted into the stratosphere by deep convection and the Brewer-Dobson circulation. The removal of aerosols from the stratosphere occurs mainly by sedimentation and through quasi-isentropic transport of air masses in tropopause folds (SPARC, 2006).

Long-term evolution of stratospheric aerosol has been a focus of several studies (see review by
Kremser et al., 2016 and references therein). Remote and in situ observations between 1970s and 2004 did not reveal any significant change in the background aerosol (Deshler et al., 2006). Several further studies (Hoffman et al., 2009; Vernier et al., 2011a; Trickl et al., 2013) reported an increase of stratospheric aerosol levels since 2002, whereas the source of this increase was debated. Initially this increase was attributed by Hoffman et al. (2009) to a rapid rise of Asian sulfur emissions,
uplifted by deep convection within the Asian monsoon. Vernier et al. (2011a) used global satellite observations to demonstrate that the increase was primarily caused by minor volcanic eruptions, whose impact should be carefully accounted for when analyzing the change in aerosol load. Although of much smaller significance compared to Pinatubo or El Chichon, these minor eruptions had a notable effect on climate (Solomon et al., 2011; Fyfe et al., 2013; Santer et al., 2014; 2015;
Andersson et al., 2015), suggesting that even small variability of stratospheric aerosol matters.

It is now widely accepted that volcanic eruptions largely determine the observed variability of stratospheric aerosol load (Kremser et al., 2016). Meanwhile, recent studies report a measurable increase of non-volcanic component of aerosol within Asian Tropopause Aerosol Layer (ATAL), occurring during Northern summer above the Asian monsoon (Vernier et al., 2015; Yu et al., 2015).
Accurate long-term measurements are indispensable to quantify the human-induced change in stratospheric aerosol.

While measurements from space are performed with a large diversity of techniques, long-term ground-based observations are highly valuable as they ensure the continuity and coherence of stratospheric aerosol record. During volcanically quiescent conditions accurate detection of
stratospheric aerosols becomes challenging as the aerosol scattering signal becomes small compared to the molecular scattering. In an effort to better characterize the evolution of stratospheric aerosol load and its variability at Northern mid-latitudes during the post-Pinatubo era we utilize a continuous 22-year observation record from Observatoire de Haute-Provence and a variety of satellite data sets.

The paper is organized as follows: Section 2 provides information on the OHP lidars, aerosol
retrieval and satellite data sets exploited. Section 3 compares the OHP lidar and satellite aerosol records. Section 4 provides examples of volcanic plumes detections and distinguishes volcanically-perturbed and quiescent periods. Section 5 describes the variability, annual cycle and long-term change of background aerosol. Section 6 discusses the proposed interpretation and concludes the paper.


## 2 Instruments and data sets
### 2.1 Observatoire de Haute-Provence lidars



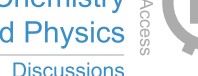

The Observatoire de Haute-Provence (OHP) located in Southern France (43.9° N, 5.7° E, 670 a.s.l.) is one of the Alpine stations within the Network for Detection of Atmospheric Composition

Change (NDACC). The site is characterized by a high rate of clear nights and offers an opportunity for frequent lidar observations.

For over three decades two independent lidar systems have been operated at OHP station: a Differential Absorption Lidar (DIAL) for stratospheric ozone (hereafter referred to as LiO3S) and a Rayleigh-Mie-Raman lidar for middle atmosphere temperature measurements (hereafter referred to

as LTA). Both LiO3S (Godin-Beekmann et al., 2003) and LTA (Hauchecorne et al., 1992) lidar systems provide routine measurements since 1985 and 1979 respectively. After a technical upgrade of both lidars in 1994 the mean measurement rate is 10-12 acquisition nights per month.

While LTA system includes a separate telescope and detection channel for clouds and aerosol (Chazette et al., 1995; Keckhut et al., 2005; Hoareau et al;., 2013), this study exploits the primary,

more powerful detection channel of LTA (532 nm) instrument, restricted to altitudes above 12-13 km due to saturation of detectors by strong Rayleigh returns below.

The off-line channel of LiO3S lidar features Nd:YAG laser frequency-tripled to 355 nm, which operates at 50 Hz pulse rate and 42 mJ/pulse energy. The total collective surface of its mosaic 4-mirror telescope is 0.88 m2. The primary low gain channel of LTA makes use of a frequency-

doubled Nd:YAG laser emitting at 532 nm with a pulse rate of 50 Hz and 350 mJ/pulse energy and a 0.03 $m^2$ telescope. The maximum vertical resolution for both lidars amounts to 15 m, however the vertical profiles are usually reported at 150 m resolution.

### 2.2 Lidar aerosol retrieval


For retrieving vertical profiles of stratospheric aerosol we use LiO3S and LTA measurements spanning 1994 through 2015 with a total number of 3118 (LiO3S) and 2691 (LTA) nights of lidar acquisitions, lasting 3-5 hours each. The retrieval is based on the Fernald-Klett inversion method (Fernald, 1984; Klett, 1985), which provides backscatter and extinction coefficients. The scattering

ratio (SR) is computed as a ratio of total (molecular plus aerosol backscattering) to molecular backscattering. The reference zero-aerosol altitude is set between 30 and 33 km. LiO3S 355 nm data are converted to 532 nm using wavelength exponents for particle extinction ($\kappa e$) and backscatter ($\kappa b$) adapted from Jäger and Deshler (2002; 2003) and set to $\kappa e=1.6$ and $\kappa b=1.3$ after the year 1997. Similarly, the extinction-to-backscatter (lidar) ratio is set to 50 sr after 1997, which is a commonly

assumed value for volcanically-quiescent conditions and periods of moderate eruptions (e.g. Trickl et al., 2013; Ridley et al., 2014; Sakai et al., 2016). The molecular backscatter is calculated from National Centers for Environmental Prediction (NCEP) daily meteorological data interpolated to OHP location. The lidar raw signals have been subjected to a thorough quality screening, accounting for the instruments' technical health log. The overall rejection rate amounted to 17% and 11% for

LiO3S and LTA respectively.

Cumulative uncertainties of the backscatter measurements induced by random detection processes, possible presence of aerosol at the reference altitude and the error in lidar ratio value do not exceed 7% as reported by Chazette et al., (1995). Another major source of uncertainty is the molecular number density derived from atmospheric pressure and temperature. The lidar inversion is

particularly sensitive to the molecular density at the reference altitude, where the lidar return is assumed to be purely due to molecular scattering. Since the routine radiosonde measurements, commonly used to derive the molecular density, rarely reach the reference altitudes above 30 km, reanalysis data are required for the inversion.

We compared the monthly-mean series of integrated backscatter coefficient in 17 - 30 km layer

retrieved using NCEP and ERA-Interim reanalyses and found a mean relative difference of 5.6 % between both datasets. This value may serve as an estimate for the uncertainty due to molecular density. As a result, the total uncertainty of individual backscatter measurement is below 10 %. We note that the uncertainty in the assumed lidar ratio has a very limited effect on the derived values of backscatter coefficient and scattering ratio (~0.15 %/sr). At the same time, error in lidar ratio affects





proportionally the aerosol extinction and optical depth, whose uncertainty may thus be somewhat larger.

### 2.3 Satellite aerosol sounders

SAGE II (Stratospheric Aerosol and Gas Experiment) (Russel and McCormick, 1989) is a seven-channel Sun photometer. It was launched onboard the Earth Radiation Budget Satellite in 1984 and provided solar occultation measurements of stratospheric aerosol extinction with a vertical resolution of 1 km until mid-2005. SAGE II fully covers the latitude range from 80° S to 80° N in 1 (2) month with a typical rate of 32 measurements per day (reduced to 16 after 2000). We used SAGE

II version 7.0 aerosol extinction data at 525 nm converted to 532 nm using a wavelength exponent $\kappa e$=1.6.

    GOMOS (Global Ozone Monitoring by Occultation of Stars) (Bertaux et al., 2010), is a UV/Visible/NIR spectrometer launched in 2002 onboard ENVISAT and operating until April 2012. The instrument performed occultations of selected stars by means of four spectrometers. We use

aerosol extinction profiles retrieved by the AerGOM algorithm which was developed using an improved aerosol parameterization (Vanhellemont et al., 2016)

    OSIRIS (Optical Spectrograph and InfraRed Imaging System) is a limb scatter instrument launched onboard the Odin satellite in 2001 and providing measurements of various chemical species and aerosol extinctions (McLinden et al., 2012). The primary instrument is Optical Spectrograph

(OS) operating in 284-810 nm range and providing between 100 and 400 occultations per day depending on the time of year. The principle of limb scattering and the Odin satellite orbit limit the coverage in the winter hemisphere in such a way that no data are available above 45° N during 2-month period around the winter solstice. We use OSIRIS version 5.07 stratospheric aerosol extinction data at 750 nm (Bourassa et al., 2012) converted to 532 nm using $\kappa e$=2.0.

CALIOP (Cloud-Aerosol Lidar with Orthogonal Polarization) onboard CALIPSO satellite platform is a nadir-viewing active sounder (Winker et al., 2010). Operational since June 2006, CALIOP provides range-resolved measurements of elastic backscatter at 532 nm and 1064 nm with a vertical resolution of around 200 m in the stratosphere. CALIOP lidar makes use of a Nd:Yag laser operating at 20.2 Hz with a 110 mJ/pulse power and a 0.78 m$^2$ telescope. The data used here are

based on night-time 532 nm level 1B version 4.00 product, post-processed using a treatment described by Vernier et al. (2009). The backscatter data are converted to extinction using lidar ratio of 50 sr and data are cloud-cleared in the upper troposphere using a depolarization ratio threshold of 5%.

    OMPS (Ozone Mapping Profiling Suite) LP (Limb Profiler) onboard NPP/Suomi satellite,

launched in 2012 measures limb-scattered light with a sampling rate of up to 7000 measurements per day (Jaross et al., 2014). Regular observations of aerosol extinction are available since April 2012. We use V0.5 extinction data at 675 nm (DeLand et al., 2016) converted to 532 nm using $\kappa e$=1.8.

### 3 Intercomparison of OHP lidars and satellites sounders


    Figure 1 shows time series of monthly-averaged stratospheric Aerosol Optical Depth between 17 and 30 km altitude (sAOD$_{1730}$) derived from OHP lidars and satellite data sets. The choice of the lower integration boundary is explained hereinafter. Monthly-mean values comprise on average of 9 (LiO3S) and 11 (LTA) individual acquisition nights (after quality screening), whereas the satellite

zonally- and monthly-averaged values contain 72 (SAGE II), 128 (GOMOS), 97 (OSIRIS), ~ 4·10$^6$ (CALIOP) and ~ 3·10$^3$ (OMPS) individual measurements. The average standard deviation for monthly averages of OHP lidars' sAOD$_{1730}$ amounts to 10.4 % (LiO3S) and 7.8% (LTA). The agreement between all data sets is remarkable despite the large variety of measurement techniques. The results of intercomparison are summarized in Tab. 1.

The OHP lidars agree to within 0.7 ± 2.4% (mean relative difference and two Standard Errors, 2 SE) with a correlation coefficient of 0.9. The LiO3S and LTA lidars compared to the satellite mean

$sAOD_{1730}$ show a difference of -1.5 ± 2.2% (2SE) and -2.7 ± 2.1% (2 SE) respectively with a correlation of 0.94 for both lidars. The satellite-to-satellite intercomparison shows mean discrepancies below 9% and correlation above 0.8 for any satellite pair except OMPS, whose observation record length is less than 4 years and covers a period with small geophysical variability. Note that the discrepancies may be partly caused by the error in the assumed wavelength exponents and lidar ratio. Indeed, the largest lidar-satellite discrepancies are obtained for the satellite sounders operating at higher wavelengths, i.e. OSIRIS (750 nm) and OMPS (675 nm), whereas the best agreement is between OHP LTA lidar and CALIOP, both operating at 532 nm.

Overall, all the biases are well below the statistical errors, which confirms the coherence between the continuous OHP record and the combined satellite time series. Note that the satellite series are zonally averaged over 10° latitude belt centred at OHP latitude in order to increase the sampling. The coherence between lidar and satellite series suggests that the stratospheric aerosol burden is zonally-uniform at least on a monthly-mean scale. This may be explained by the presence of strong zonal winds in the stratosphere, which rapidly homogenize the aerosol and tracers in the zonal direction.

The layer between 17 and 30 km, for which the comparison is reported in Fig. 1 and Tab. 1 does not represent the total stratospheric aerosol column. A significant fraction of stratospheric aerosol resides below 17 km (Rideley et al., 2014; Andersson et al., 2015), however an accurate detection of the aerosol abundance in the lowermost stratosphere is more challenging for limb-viewing satellite instruments (Bourassa et al., 2010; Thomason and Vernier, 2013), which may lead to larger discrepancies in sAOD.

Figure 2 displays a comparison of aerosol extinction profiles averaged over two 20-month volcanically-quiescent periods 2002-2003 and 2013-2014 covered by time-overlapping observations by two different triplets of satellite sounders. The comparison reveals close agreement between all data sets in the 17-25 km layer whereas above 25 km the lidar profiles show some negative bias compared to satellites, which is most likely related to an error in lidar calibration, relying on the assumption of the absence of aerosol above 30 km, which – as suggested by CALIOP data calibrated at higher altitudes - may not always be the case.

Below 17 km the extinction profiles diverge and both OHP lidars show higher values compared to satellite ones. This discrepancy may be due to the use of fixed lidar ratio and wavelength exponents, which may vary with height depending on the size distribution of aerosol. Particularly, the lower stratospheric layers are expected to contain larger particles, accumulating after sedimentation from the higher stratospheric levels (SPARC, 2006). Throughout the paper we restrict to aerosol measurements above 15 km because the lower layer may be contaminated by cirrus clouds, which according to Goldfarb et al. (2001) and Hoareau et al. (2013) may encounter at altitudes up to 14 km above OHP. In addition, measurements by LTA system (optimized for the middle atmosphere) may be affected by incomplete desaturation of the strong lidar returns from lower layers.

## 4 Detection of volcanic plumes and quiescent periods

The remarkable coherence between the lidar- and satellite-based $sAOD_{1730}$ series demonstrated in the previous section allows for a synergetic use of local and global observations to characterize at best the variability of stratospheric aerosol. Fig. 3 shows $sAOD_{1730}$ series computed by averaging the OHP lidars and all five satellites data sets. The timing of VEI=4 volcanic eruptions North of 20° S is indicated by vertical arrows, whereas the periods affected by these eruptions are marked by light blue shading (selection criteria are described hereinafter).

*Quiescent period 1997 - 2003*

The $sAOD_{1730}$ series since 1994 shows a tail of Pinatubo aerosol followed by a stabilisation at a quasi-constant level around mid 1997 according to SAGE II and OHP lidars observations. Between mid-1997 and late 2001 aerosol loading remains stable with no discernible eruption-induced enhancements at Northern Hemisphere (NH) mid-latitudes. This is fully consistent with other mid-


latitude lidar observations (Deshler et al., 2006; Trickl et al., 2013). Although some VEI 4 eruptions between 2000 and 2003 have occurred over that time, they had very limited stratospheric impact (Vernier et al., 2011a; Kremser et al., 2016).

Importantly, the stratospheric aerosol levels during 1997-2003 period are at or below any previous background period since 1970 (Jäger, 2005; Deshler et al., 2006) and may thus be regarded as a reference level for background stratospheric aerosol, against which further changes in aerosol load should be compared. Both SAGE II and OHP lidars report an average background $sAOD_{1730}$ for the "reference" quiescent period of $2.3 \cdot 10^{-3} \pm 2.4\%$ (2 SE), which is marked in Fig. 3 by dashed line and grey shading, indicating 1-$\sigma$ range of values.

### *Volcanically-active period 2003-2013*

The continuous quiescent period is terminated in late 2003, when the plume of tropical Ruang and Reventador eruptions (Thomason et al., 2008) reaches NH mid-latitudes. The subsequent VEI=4 eruptions of Manam volcano at 4 °S (Vanhellemont et al., 2010), Soufriere Hills at 16° N (Prata et al., 2007) and Tavurvur at 4° S lead to step-like increases of $sAOD_{1730}$. In Summer 2008, two neighbouring VEI 4 eruptions of Okmok and Kasatochi volcanoes at 55° N (Bourassa et al., 2010) result in a rapid increase of $sAOD_{1730}$ followed by a relaxation to quasi-background level with e-folding time of 6 months. In June 2009, the eruption of Sarychev at 48° N (Haywood et al., 2010) increases $sAOD_{1730}$ to $8 \cdot 10^{-3}$, the highest value since 1994. The post-Sarychev recovery is relatively fast, with only 4-5 months of e-folding period, after which $sAOD_{1730}$ returns to background level in January-February 2010.

A strong enhancement of $sAOD_{1730}$ follows the eruption of Nabro volcano (14° N) in June 2011. A rapid hemisphere-wide dispersion of Nabro plume was facilitated by the Asian monsoon (Bourassa et al., 2012; Fairlie et al., 2014), although the role of the monsoon in providing an alternative pathway for aerosol and/or $SO_2$ into the stratosphere is debated (Vernier et al. 2013). Interestingly, the mid-latitude Sarychev eruption and the tropical Nabro eruption resulted in $sAOD_{1730}$ enhancement of nearly the same amplitude, however the removal of Nabro aerosol took much longer time (e-folding period of up to 19 months) according to zonal-mean series derived from CALIOP and OSIRIS.

A better insight into the temporal evolution and vertical structure of Sarychev and Nabro plumes is provided by Fig. 4, showing scattering ratio (SR) profiles obtained by OHP LiO3S lidar during the corresponding volcanic periods. The Sarychev plume was detected at OHP already 10 days after the eruption as sharp SR enhancements below 19 km reaching 3.4 units. A remarkable scatter between the individual profiles points to a rapid three-dimensional evolution of the plume, dispersed by the stratospheric mean zonal flow, which reversed over the course of the plume permanence.

The first signatures of Nabro plume were detected at OHP 45 days after the start of eruption. In contrast to the highly variable Sarychev plume, the SR profiles bearing Nabro signature are characterized by a smooth broad-range enhancement reaching 1.7 and relatively small scatter between the individual profiles, suggesting that the plume has already been mixed with the ambient air before arriving at OHP latitude. Noteworthy, the period-averaged SR profiles (black circles in Fig. 4) corresponding to Sarychev and Nabro plumes are of comparable shape and enhancement amplitude.

### *Post-Nabro period*

By early-mid 2013 all data sets report $sAOD_{1730}$ stabilizing around $2.7 \cdot 10^{-3} \pm 3.9\%$ (mean lidars) and $3.0 \cdot 10^{-3} \pm 3.5\%$ (mean satellites) and remaining at that level for almost 2 years, through December 2014. In January 2015, the plume of Kelud eruption (Kristiansen et al., 2015) reached OHP latitude as inferred from CALIOP $sAOD_{1730}$ time-latitude section (not shown). The signatures of Kelud plume were observed at OHP until April 2015, after which $sAOD_{1730}$ returns to near background levels. The eruption of Calbuco volcano at 41 °S (Romero et al., 2016) in spring 2015 has led to a notable increase of stratospheric aerosol load, as suggested by CALIOP (Fig. 6) and OMPS (not shown) observations, which however provide no indications for the transport of Calbuco





plume to OHP latitude in 2015. Initial analysis of the more recent OMPS data by does not indicate significant transport of the Calbuco plume into the Northern Hemisphere through July 2016 (G. Taha, personal communication).

**4.1 Selection of volcanically-perturbed periods**

Since 1994 the major perturbations of NH stratospheric aerosol load were caused by mid-latitude volcanic eruptions of Okmok/Kasatochi and Sarychev as well as the tropical eruption of Nabro, all readily reflected in OHP lidars and satellite $sAOD_{1730}$ series (Fig. 1 and 3). The plumes

of more distant (tropical) eruptions are not always obvious in sAOD series. In order to accurately distinguish between volcanically-perturbed and quiescent periods we use global-coverage satellite observations to track the spatiotemporal evolution of each volcanic plume. The plumes were detected by examining time-latitude sections of partial sAOD from all satellite records (example for CALIOP is provided in Fig. 6). If a plume was found to extend beyond the tropical belt towards the

Northern extra-tropics, the temporal extent of the corresponding volcanic period was determined by comparing the OHP lidar monthly-mean $sAOD_{1730}$ values and SR profiles obtained after an eruption against those averaged over the "reference" quiescent period 1997-2003. In this way, a period is considered as volcanically-perturbed if both of the following two conditions are fulfilled:

i) monthly-mean $sAOD_{1730}$ value exceeds the 1-σ range of the "reference" quiescent period of

1997-2003 (grey band in Fig. 3);

ii) monthly-mean SR profile exceeds the 1-σ range of the "background" SR profile - an average over the entire "reference" quiescent period (grey-filled in Fig.5) in a layer > 2 km thick.

Figure 5 shows the difference between averaged SR profiles for the quiescent and volcanically-perturbed periods in order to clarify the application of the second selection criterion.

The black solid curve and grey shading represent the mean SR profile for the "reference" period (1997-2003) and its 1-σ range respectively. The coloured curves show SR profiles corresponding to the aged plumes of tropical eruptions of Ruang, Nabro and Kelud. The maximum SR values of these profiles are remarkably smaller than those observed in a young plume (Fig. 4), however they are visibly beyond the grey-shaded background range of SR. The same consideration holds for the

corresponding $sAOD_{1730}$ values in Fig. 3. This allows for classifying the respective periods as volcanically-perturbed. The timing of VEI 4 eruptions and the lifetime of their plumes as detected at OHP are listed in Tab. 2.

We noted that the time required for a plume to propagate to OHP latitude depends on the eruption season and injection altitude. In particular, the tropical eruptions injecting material directly

into the lower stratosphere (e.g. Soufriere Hills or Kelud) would have a longer lifetime in the stratosphere, however their poleward propagation is inhibited during Boreal summer, when stratospheric meridional exchange weakens. For this reason, the Kelud plume has reached OHP latitude only about 10 months after the eruption. The period between the full decay of Nabro plume in early 2013 and the arrival of the aged Kelud plume in late 2014 is characterized by an SR profile

(dashed curve in Fig. 5) lying within the background range of values. The $sAOD_{1730}$ is relatively stable and remains within the background range during this period, which is therefore classified as quiescent.

Another example is the eruption of Merapi (7° S), which occurred in October 2010, shortly before the start of the NH winter season, characterized by enhanced poleward transport into the

winter hemisphere. According to CALIOP observations (Fig. 6), the plume of Merapi was transported to OHP latitude in about 2 months, whereas its permanence at NH mid-latitude was limited to 3 months.

**5 Non-volcanic drivers of aerosol variability**


Figure 6 displays the time-latitude section of zonal-mean AOD in a layer between 15 and 19 km ($sAOD_{1519}$) from CALIOP data and time series of the same quantity obtained by OHP LiO3S





lidar. The 15-19 km layer is chosen because it is directly impacted by most of VEI 4 eruptions and is characterized by efficient quasi-isentropic exchange within the UT/LS (e.g. Kremser et al., 2016). The enhanced poleward transport into the winter hemisphere is exhibited by meridional wind vectors in Fig. 6.

Beside the volcanic plumes, CALIOP observations reveal systematic enhancement in $sAOD_{1519}$ between about 15° and 45° N during the Northern summer, most prominent ones occurring in 2007, 2010, 2013 and 2015. Given its timing and location, this feature can be attributed to the so called Asian Tropopause Aerosol Layer (ATAL) (Vernier et al., 2011b; Thomason and Vernier 2013), occurring in the 15-18 km layer above the Asian summer monsoon and extending to mid-latitudes (Vernier et al., 2015).

Another feature revealed by CALIOP is systematic aerosol depletion in January-February around the equator and spreading poleward. The tongues of aerosol-poor air are readily discernible in 2007, 2008, 2010, 2012 and 2015 whereas in the other years they are scrambled by volcanic plumes or hardly discernible from the low background aerosol burden. The timescale of poleward transport of clean air can be inferred from the shape of the clean air tongues – fast within the tropical belt and slower across the subtropical stratospheric barrier. The systematic aerosol depletion in the TTL during Austral summer was attributed by Vernier et al. (2011c) to fast convective cross-tropopause transport (overshooting) of clean tropospheric air (cleansing). The clean air reaches OHP latitude in about 3 months, which is reflected in the OHP lidar series, showing a recurring minimum in late spring - early summer.

The time-latitude pattern of $sAOD_{1519}$ can be paralleled with that of water vapour at 100 hPa level from Aura Microwave Limb Spectrometer (MLS) (Waters et al., 2006) version 4.2 data (Livesey et al., 2015). Dashed and dotted contours in Fig. 6 encircle the areas of water vapour mixing ratio of 3, 4 and 5 ppmv. The 5 ppmv (red dashed) contour shows the area of annual maximum of water, emerging during the Northern summer, which can be attributed to the moisture flux from the Asian monsoon (Park et al., 2007; Schwartz et al., 2015). The moist air is traceable to OHP latitude and coincides in time and space with the annual maximum of $sAOD_{1519}$, associated with ATAL. Spatiotemporal match of the aerosol and water vapour annual maxima suggests the same origin of the both – the Asian monsoon.

The areas of annual minimum of water vapour (black dashed contours) correlate with the minima in TTL aerosol load, both occurring during the Southern summer. While the annual minimum of water vapour can be readily explained by the coldest TTL temperatures in January-February leading to enhanced dehydration of the TTL (e.g. Holton et al., 1995), the aerosol reduction can be attributed to convective cleansing during Austral summer (Vernier et al., 2011c). Both dry and clean air features show similar poleward propagation. Overall, the seasonal cycle of stratospheric aerosol loading in the TTL, featuring a maximum in NH during Boreal summer and minimum around the equator during Austral summer is similar to that of water vapour.

### 5.1 Annual cycle

Fig. 7a shows climatological annual cycle of scattering ratio (SR) profile from OHP LiO3S lidar based on the selected periods considered as volcanically-quiescent (see Fig. 3). Throughout the seasons and altitude layers the SR does not exceed 1.08, meaning that for the quiescent conditions the aerosol backscatter constitutes less than 8% of the molecular backscatter. The permanent layer of aerosol in the stratosphere, also referred to as Junge layer (Junge et al., 1961), is commonly attributed to sulphuric gas precursors emitted at the surface and eventually transformed into $H_2SO_4$-$H_2O$ liquid aerosol mixture (e.g. Brock et al., 1995).

The amplitude of annual cycle of background aerosol is small but variable with altitude. The upper boundary of Junge layer peaks in winter, which is likely related to a weaker transport barrier between the tropical aerosol reservoir and mid-latitude stratospheric overworld during Northern winter, when the wave induced meridional mixing in NH is most pronounced (Holton et al., 1990; Hitchman et al., 1994). Note that the meridional divergence of tropical air in the stratosphere is also



modulated by the QBO, where the westerly shear phase favours the poleward transport during
       northern winter (Trepte and Hitchman, 1992).
           In the middle layer (19-25 km), the SR varies between 1.04 and 1.07 and shows a smooth
       maximum in Spring. The lowermost layer, below 19 km exhibits a more pronounced annual cycle,
       featuring a minimum in May at 16 km, which propagates to 17 km by the end of August. In view of
its altitude range and timing, this minimum can be attributed to advection of convectively-cleansed
       air from the TTL after the Austral summer convective season (Vernier et al., 2011c) and reaching
       mid-latitudes in about 3 months as was concluded from Fig. 6. The late spring minimum appears to
       be a robust feature captured by all other satellites (not shown), independently of the observation
       period. Starting from August, the clean air in the LS is progressively replaced by aerosol-enriched air,
presumably originating from the ATAL. Note that the initial inference on the extension of ATAL to
       OHP latitude is made on the base of CALIOP time-latitude section in Fig. 6.  The SR between 15
       and 16 km reaches a maximum in October and reduces gradually over the course of the winter.
       Importantly, for any quiescent subperiod over the course of 22 yr OHP series, the pattern is
       essentially the same.
430        Fig 7b provides a satellite zonal-mean view on the non-volcanic aerosol annual cycle observed
       by CALIOP since 2006. The month-altitude pattern of zonal-mean background aerosol revealed by
       CALIOP is fairly similar to that obtained by OHP lidar. The main features, namely the winter
       maximum of the Junge layer upper boundary, the spring maximum of SR in the middle layer (19-25
       km) and the upward propagation of the late-spring clean feature are readily discernible in both OHP
and CALIOP climatologies. Whereas the general patterns appear similar, comparison of the SR
       vertical distribution reveals some discrepancies between the OHP lidar and CALIOP, with the latter
       showing slightly higher (lower) SR values in the uppermost (lowermost) layer. The same conclusion
       was drawn from the comparison of extinction profiles in Fig. 2 (see Sect. 3 for interpretation of the
       discrepancies). In addition, the maximum of SR in autumn at 15-16 km, attributed to ATAL, is less
pronounced in CALIOP section, which is likely due to zonal averaging, reducing the meridionally-
       restricted ATAL signal (Fig. 2 in Vernier et al., 2015).
           In the previous section we noted a relation between time-latitude variation of aerosol and water
       vapour in the lower stratosphere. Fig. 7c provides further evidence to this finding.  Similarly to
       aerosol, the LS water vapor annual cycle exhibits the upward propagation of the late-spring
minimum, followed by the maximum in autumn. As already pointed out on the base of Fig. 6, both
       aerosol and water vapour in the mid-latitude LS are modulated by poleward transport of clean (dry)
       air from the deep tropics and aerosol-rich (wet) air from the Asian monsoon region. In fact, the
       annual cycle the extra-tropical water vapour bears an imprint of the tropical $H_2O$ "tape recorder"
       (Mote et al., 1996) lagged by the timescale of poleward transport from the TTL  (e.g. Hoor et al.,
2010). The same applies effectively to background aerosol, leading to similar month-altitude patterns
       of aerosol and water, as Fig. 7 suggests.

       **5.2 Long-term change in stratospheric aerosol burden**

455        Detection of long-term change in non-volcanic component of stratospheric aerosol is
       complicated by frequent minor eruptions of stratovolcanoes, whose plumes may persist in the
       stratosphere for several years, whilst decaying exponentially. A thorough analysis of the trends in the
       background stratospheric aerosol over 1971-2004 period (covering 3 quiescent periods) was carried
       out by Deshler et al. (2006), who concluded on the absence of long-term change.  The 22-year
stratospheric aerosol series provided here covers two quiescent periods: the "reference" six-year long
       period 1997-2003 and a recent post-Nabro two-year long period 2013-2014. This new-era quiescent
       period is characterized by stabilization of stratospheric aerosol load at near-background level,
       rendering it suitable for comparison against the "reference" quiescent period. In this way, a positive
       change of 13.9 ± 4.5% (2SE) can be inferred by comparing average $sAOD_{1730}$ values over the two
periods. This estimate may be considered as an upper limit of the trend in non-volcanic aerosol the
       NH mid-latitude stratosphere, however not without caution. First, it is the limited time span of the





new quiescent period, three times shorter than the "reference" one. Second, a possible influence of eruptions with VIE=3, which may occasionally penetrate into the stratosphere (Carn et al., 2015; Mills et al., 2016). The second, however, may as well be true (although not detected or reported) for the "reference" quiescent period. Furthermore, the observations exploited here provide no indication of the influence of eruptions other than those listed above (Tab. 2).


If the change in stratospheric aerosol load is largely due to non-volcanic processes, then the most likely source is the growing Asian emissions of aerosol precursors (Smith et al., 2011), transported into the lower stratosphere by the Asian monsoon (Randel et al., 2010). Indeed, the AOD of ATAL over Eastern Mediterranean, downwind of South-East Asia (Lawrence and Lelieveld, 2010), has increased three times since the late 1990s as inferred from SAGE II and CALIOP observations by Vernier et al. (2015). OHP site is influenced by the Asian anticyclone and its composition, as shown above, hence the change in ATAL AOD is expected to be reflected in OHP long-term series. However, given that the manifestation of ATAL signal in OHP observations is limited to autumn season and lower stratosphere, the change in non-volcanic aerosol should be evaluated with respect to the season and the layer.



Fig. 8a displays vertically and seasonally resolved change in non-volcanic sAOD over 18 years. The statistically significant increase by a factor of two in LS is restricted to late summer and early fall, i.e. in phase with the Asian monsoon signatures detected in aerosol and water vapor. Note that little or no (statistically significant) increase is observed in other seasons, which suggests that accumulation of volcanic aerosols (if any) is unlikely to be the reason for the positive trend. Indeed, zero change in the LS during late spring, i.e. when the tropical air reaches NH mid-latitudes, rules out the effect of unaccounted tropical plumes on the trend estimates.


Further insight into the long-term change of background aerosol is provided in Fig 8b, showing the evolution of AOD in September within the altitude layer characterized by the maximum growth of AOD. Both OHP lidar and satellites provide a clear indication of the increase of AOD with time. The value in 2010, representing the post-Sarychev quiescent period, is slightly higher than the post-Nabro values, however its contribution to the linear regression is limited to 12 %. The linear regression essentially rests upon the two quiescent periods separated in time: 1998 – 2004 and 2013 – 2015, hence the trend value largely depends on the quantification of the aerosol level during the second period. This post-Nabro quiescent period was interrupted by the arrival of Kelud plume at OHP latitude in early 2015. By September 2015 the Kelud plume is no longer observed at OHP: the value in September 2015 is not much different from the pre-Kelud observations in 2013 and 2014, which suggests that the trend estimate is unaffected by the Kelud plume. Lidar observations at Tsukuba, 36° N (Sakai et al., 2016) do not show indication of the presence of Kelud plume in 2015.




## 6 Discussion and summary


Over the last two decades NH stratosphere was perturbed by a series of minor volcanic eruptions, leaving strong but transient signals in stratospheric aerosol load. A combination of concurrent local and global observations was used to carefully separate between volcanically-perturbed and quiescent periods. The volcanic plumes and their meridional dispersion were detected using satellite observations, whereas determination of a plume's lifetime was done by comparing OHP lidar measurements against the "reference" levels of background aerosol, corresponding to 1997-2003 period. This approach suffers from the limited sensitivity of remote sensing techniques to low aerosol concentrations, however it is the best that can be provided using the available observations.


The selection of quiescent periods is particularly challenging during 2003-2012 period, characterized by frequent minor eruptions, occurring sometimes before the previous plume has fully decayed. However, the criteria applied allow identifying several brief sub-periods over 2003-2012, during which the stratospheric aerosol attains background levels. The quiescent periods, constituting




a considerable fraction (57%) of the 22-year span of OHP observations, yield a wealth of data for establishing a robust climatology of background aerosol at northern mid-latitudes.

Analysis of non-volcanic fraction of data suggests that the annual cycle of mid-latitude background stratospheric aerosol is largely driven by remote (tropical) processes: convective cross-tropopause transport of *clean* air (Vernier et al., 2011c) during southern summer and *polluted* air from the Asian monsoon (Randel et al, 2010; Vernier et al. 2015; Yu et al., 2015) during northern

summer, both followed by poleward transport. Although this interpretation is rather robust, alternative contributors should also be considered.

Alternatively, the late-spring minimum in the lower stratosphere might be attributed to release of clean air from within the Arctic vortex after its breakup or gravitational settling of larger particles and their sink through the tropopause folds (SPARC, 2006). However the time-latitude variation of

aerosol and water vapour unequivocally point to the poleward transport, thereby providing no support to these hypotheses. The clean air obviously originates from the TTL and whatever mechanisms are responsible (injections into the stratosphere or scavenging in tropopause clouds), the TTL cleansing is an important driver of the annual cycle of stratospheric aerosol at global scale. It also appears that the cleansing process not only modulates the background aerosol but limits the

lifetime of weak plumes residing mainly in the lower stratosphere.

The late-summer aerosol maximum might partly be due to mid-latitude summertime forest fires and pyroconvection, whose stratospheric impact is recognized (Fromm et al., 2008; 2010). However these events are rare and thus unlikely to contribute significantly to the multi-year averages. The coincidence between water vapor and non-volcanic aerosol annual maxima in the NH

midlatitude LS suggests that these air masses originate from the Asian monsoon, whose influence on the extratropical LS in late summer and early fall is well known (Vogel et al., 2014; Müller et al., 2016). Indeed, according to trajectory analyses by Garny and Randel (2016), 15% of the diabatic trajectories released at 360 K within the Asian anticyclone travel to the extratropical LS in 30 days or more, which is consistent with 1-2 months lag of the aerosol and water vapour maxima with respect

to the Asian monsoon season.

The influence of Asian monsoon on the composition of lower stratosphere at OHP – as suggested by our analysis – implies that the increase in ATAL AOD reported by Vernier et al. (2015) and Yu et al. (2015) should also be reflected in OHP lidar observations. Indeed, after removal of volcanically-perturbed data we observe a doubling of LS partial AOD since 1998 in late summer and

early fall, i.e. in phase with the ATAL signal detected at OHP.

Our trend estimate is consistent with that of Vernier et al (2015), who found a tripling of aerosol extinction anomaly (summer-to-winter ratio) above the Eastern Mediterranean. As it appears, the analysis of long-term change in non-volcanic aerosol with respect to the season and altitude layer is the only way to obtain a credible trend estimate, in which the effect of unaccounted volcanic

plumes is minimized. In this way, the post-Nabro quiescent period, largely determining the observed trend, provides an accurate reference for assessment of long-term change in non-volcanic aerosol load.

The annual cycle of background aerosol is shown to reflect the meridional exchange processes, whereas its long-term evolution points to increasing anthropogenic contribution to stratospheric

aerosol budget. This effect appears very small compared to volcanic influence, however it should not be ignored. Long-term continuous observations of stratospheric aerosol available from NDACC lidar network are indispensable to follow the evolution of stratospheric aerosol and detect its human-induced change.

**Acknowledgements**

All data sets and codes used to produce this study can be obtained by contacting Sergey Khaykin (sergey.khaykin@latmos.ipsl.fr). The GOMOS AerGOM data can be obtained by contacting Christine Bingen (Christine.Bingen@aeronomie.be). We thank the personnel of OHP for conducting lidar measurements. The work was done with the support of French Institut National des

Sciences de l'Univers (INSU) of the Centre National de la Recherche Scientifique (CNRS) and of





Centre National d'Etudes Spatiales (CNES). We thank Laurent Blanot (Acri ST) and Nickolay Kadygrov (IPSL) for their help with satellite data handling. OMPS LP Version 0.5 aerosol extinction coefficient data are produced by the LP processing team (https://ozoneaq.gsfc.nasa.gov/data/omps/). The AerGom project was financed by the European Space Agency (contract number 22022/OP/I-OL).
Charles Robert's research was supported by a Marie Curie Career Integration Grant within the 7th European Community Framework Programme under grant agreement n°293560, the European Space Agency within the Aerosol_CCI project of the Climate Change Initiative and the Belgian Space Science Office (BELSPO) through the Chercheur Supplémentaire programme. The following satellite    data    used    in    this    study    are    publically    available:    CALIPSO,
https://eosweb.larc.nasa.gov/project/calipso/calipso_table; SAGE II, https://eosweb.larc. nasa.gov/project/sage2/sage2_table; OSIRIS, http://odin-osiris.usask.ca/; MLS, http://mls.jpl.nasa.gov/products/h2o_product.php.

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




| $\Delta_{mean} \pm 2SE$, % | LiO3S | LTA | SAGE II | GOMOS | OSIRIS | CALIOP | OMPS | Sat_mean |
|---|---|---|---|---|---|---|---|---|
| LiO3S | | 0.65 ± 2.4 | 2.1 ± 3.3 | 0.4 ± 4.0 | -6.6 ± 3.4 | -2.9 ± 3.2 | -4.8 ± 4.3 | -1.5 ± 2.2 |
| LTA | | | -2.1 ± 4.4 | -2.9 ± 4.3 | -7.4 ± 2.7 | -0.4 ± 2.5 | -8.6 ± 3.3 | -2.7 ± 2.1 |
| SAGE II | | | | -0.1 ± 5.9 | 7.7 ± 6.0 | - | - | 2.1 ± 2.7 |
| GOMOS | | | | | -5.8 ± 3.4 | -1.6 ± 3.7 | - | -1.9 ± 1.9 |
| OSIRIS | | | | | | 7.7 ± 2.1 | 6.6 ± 4.0 | 3.2 ± 1.3 |
| CALIOP | | | | | | | -5.5 ± 2.7 | -3.1 ± 1.2 |

| R correl | LiO3S | LTA | SAGE II | GOMOS | OSIRIS | CALIOP | OMPS | Sat_mean |
|---|---|---|---|---|---|---|---|---|
| LiO3S | | 0.9 | 0.97 | 0.9 | 0.81 | 0.85 | 0.63 | 0.94 |
| LTA | | | 0.96 | 0.86 | 0.9 | 0.93 | 0.66 | 0.94 |
| SAGE II | | | | 0.7 | 0.85 | - | - | |
| GOMOS | | | | | 0.86 | 0.88 | - | |
| OSIRIS | | | | | | 0.93 | 0.65 | |
| CALIOP | | | | | | | 0.71 | |

**Table 1.** Intercomparison of stratospheric Aerosol Optical Depth between 17 and 30 km (sAOD$_{1730}$) series displayed in Fig. 1. Mean relative difference $\Delta_{mean} \pm 2$ standard errors (top) and correlation coefficient R (bottom).


| Volcano (VEI =4) | Eruption date | Latitude | Start of period | End of period |
|---|---|---|---|---|
| Rabaul (Ra) | September 1994 | 4°S | October 1994 | Undefined |
| Ulawun (Ul) | September 2000 | 5°S | Undetected | Undetected |
| Shiveluch (Sh) | May 2001 | 56°N | Undetected | Undetected |
| Ruang (Ru) | September 2002 | 2°N | November 2003 | February 2004 |
| Reventador (Re) | November 2002 | 0°N | November 2003 | February 2004 |
| Manam (Ma) | January 2005 | 4°S | April 2005 | February 2006 |
| Soufrière Hills (So) | May 2006 | 16°N | August 2006 | Undefined |
| Tavurvur (Ta) | October 2006 | 4°S | Undefined | February 2008 |
| Okmok (Ok) | July 2008 | 55°N | August 2008 | January 2009 |
| Kasatochi (Ka) | August 2008 | 55°N | August2008 | January 2009 |
| Sarychev (Sa) | June 2009 | 48°N | June 2009 | December 2009 |
| Merapi (Me) | October 2010 | 7°S | December 2010 | February 2011 |
| Nabro (Na) | June 2011 | 13°N | July 2011 | February 2013 |
| Kelud (Ke) | February 2014 | 8°S | January 2015 | April 2015 |


**Table 2.** List of volcanic eruptions of Volcanic Explosivity Index VEI=4 occurring in the tropics and Northern hemisphere (>20°S) between 1994 and 2016 as reported by Smithsonian Institution Global Volcanism Program (http://volcano.si.edu). Temporal extent of the volcanically-perturbed period





from the corresponding eruption is provided in the rightmost two columns (see Fig. 3 and text for
       detail).

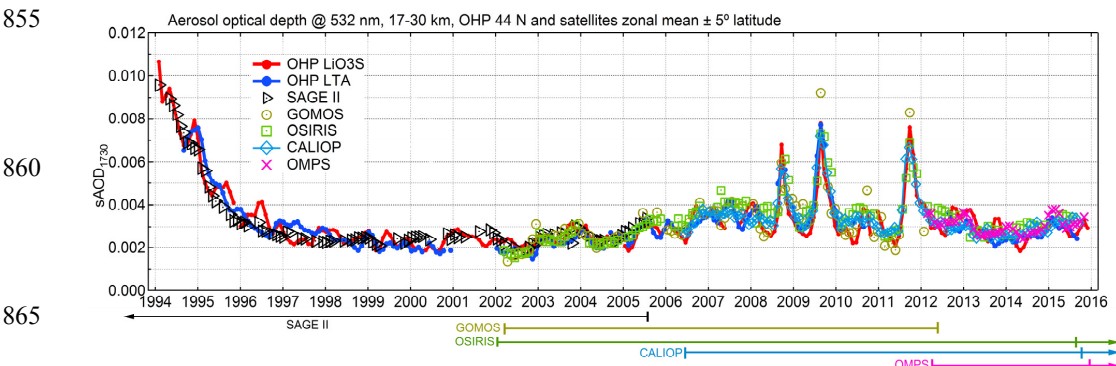




**Figure 1.** Time series of monthly-mean $sAOD_{1730}$ from OHP lidars and monthly/zonal mean
$sAOD_{1730}$ within 40°- 50° N from satellite sounders. Average statistical error of monthly-mean
$sAOD_{1730}$ from OHP lidars is provided in the embedded panel as error bars (two standard errors)
       scaled to the principal vertical axis. Time spans and data availability of satellite missions are shown
       below the panel.


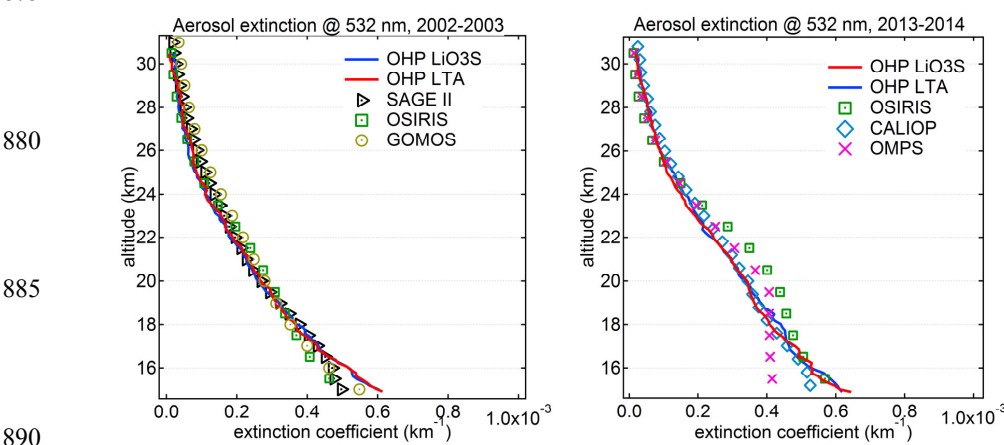



**Figure 2.** Comparison of aerosol extinction profiles at 532 nm from OHP lidars and satellites
       averaged over volcanically-quiescent periods 2002-2003 (left) and 2013-2014 (right).







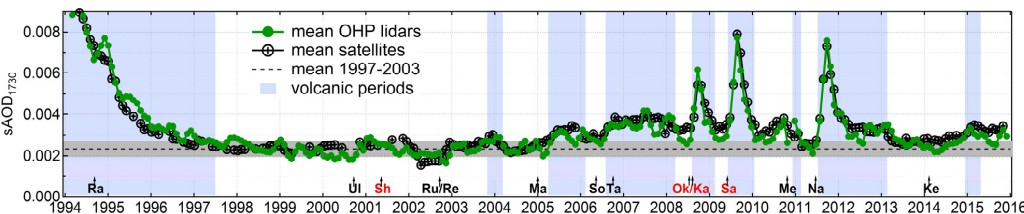

**Figure 3. Time series of monthly-mean sAOD$_{1730}$ computed by averaging both OHP lidars and all satellites. VEI 4 eruptions >20°S (Tab. 2) are indicated along the x-axis, NH mid-latitude eruptions are marked in red. Time periods considered as perturbed by volcanism are shaded light blue. See text for details.**

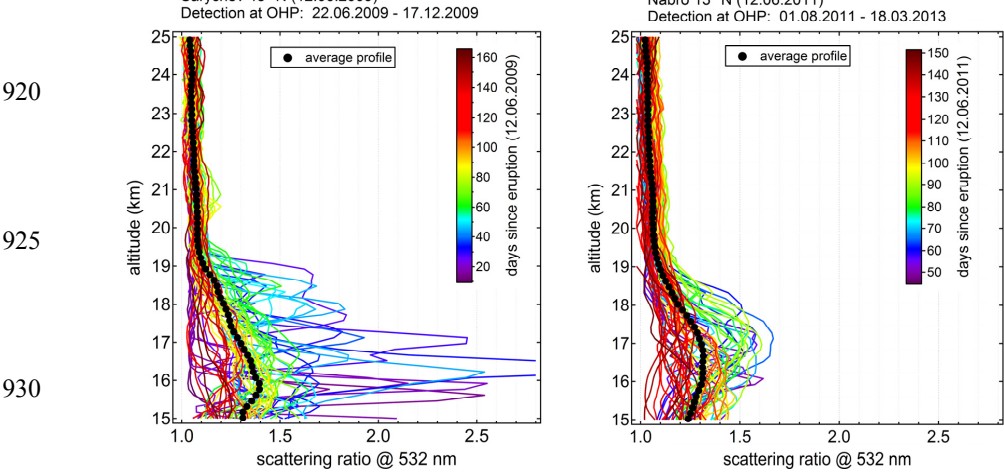

**Figure 4. Individual (coloured curves) and period-averaged (black circles) scattering ratio profiles from OHP LiO3S lidar acquired after the eruptions of Sarychev (left) and Nabro (right) volcanoes. The colours of individual profiles denote the days since eruption. The eruption dates and plume detection periods are indicated in each panel.**









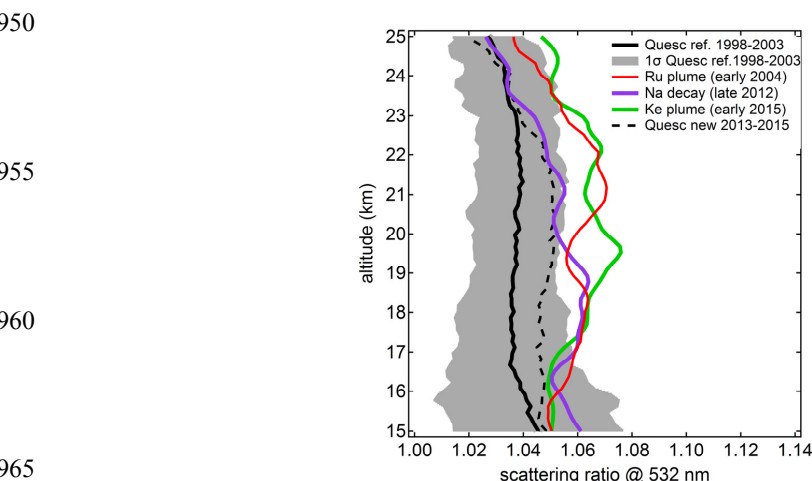

**Figure 5. Vertical profiles of scattering ratio (532 nm) averaged over different periods: "reference" quiescent period (Quesc ref. 1997-2003) and its one standard deviation range (1σ Quesc. ref.); aged volcanic plumes of Ruang/Reventador (red), Nabro in late 2012 (violet); Kelud (green); post-Nabro quiescent period (Quesc new, black dashed). See Fig. 3 and Tab. 2 for detail on period definition.**


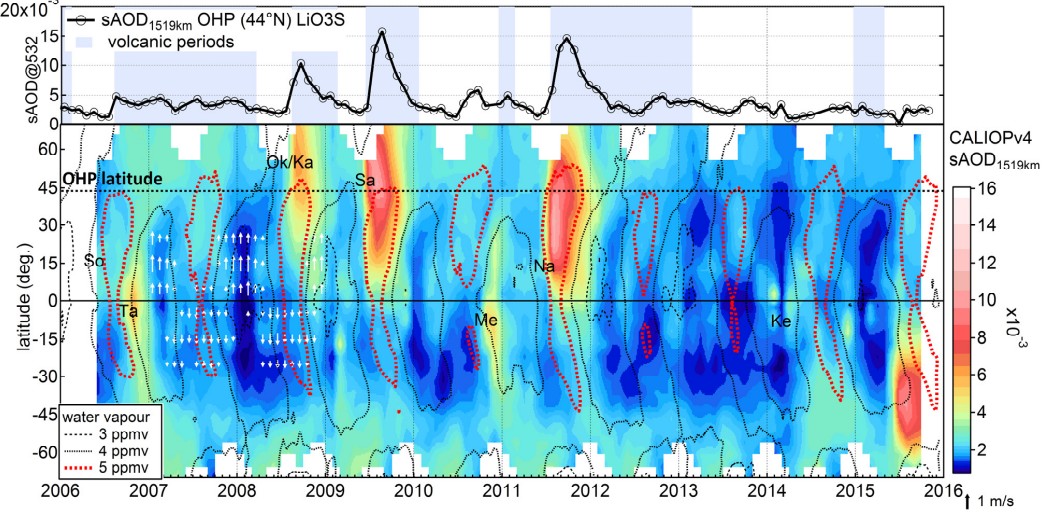


**Figure 6. Time series of monthly-mean sAOD$_{1519}$ from OHP LiO3S lidar (top) and time-latitude section of sAOD$_{1519}$ from CALIOP in log-scaled color map with indications of VEI 4 eruptions (bottom). Time periods considered as perturbed by volcanism (Tab. 2) are shaded light blue in the top panel. White arrows (in 2007-2008) represent the mean meridional component of monthly/zonally-averaged horizontal wind at 100 hPa from ERA-Interim reanalysis. Dashed and dotted contours depict zonal-mean water vapour mixing ratio at 100 hPa from Aura MLS.**














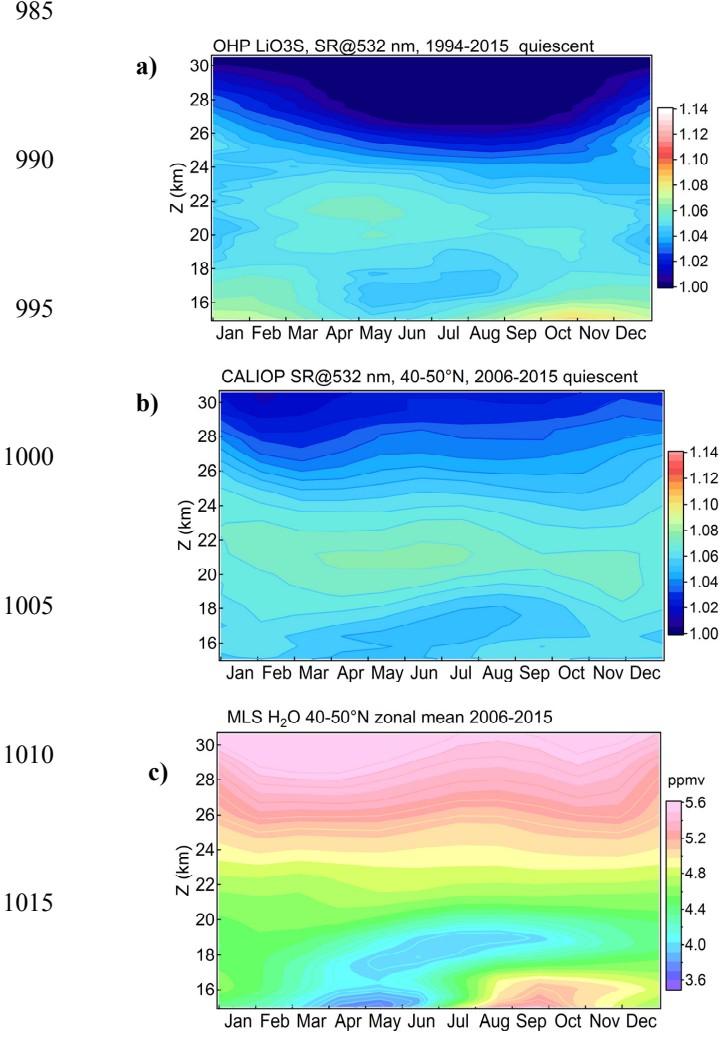

**Figure 7. Climatological month-altitude sections of a) SR from OHP LiO3S lidar for selected volcanically-quiescent periods over the entire measurement time span (1994-2015); b) zonal-mean SR at 40°-50° N from CALIOP, June 2006 - September 2015 for selected volcanically-quiescent periods (Tab. 2); c) zonal mean water vapour at 40°-50° N from MLS, June 2006 - September 2015.**









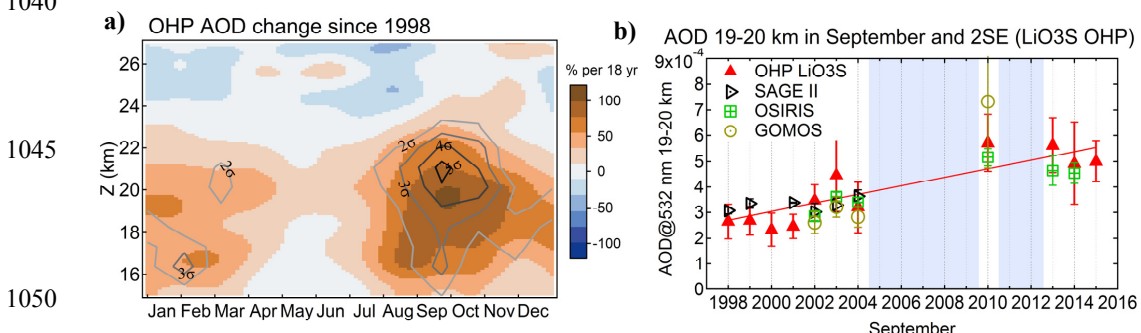

**Figure 8. (a) Monthly-averaged 1-km AOD change since 1998 from OHP LiO3S lidar based on the observations during volcanically-unperturbed periods. Statistically significant changes above 95% confidence interval are encircled by grey-scaled contours. (b) Evolution of the AOD in the 19-20 km layer in September from OHP LiO3S lidar and satellite observations above Western Mediterranean. Error bars denote two times the standard error. Shaded areas indicate the volcanically-perturbed periods.**