# Peer review of "Variability and evolution of mid-latitude stratospheric aerosol budget from 22 years of ground-based lidar and satellite observations"

_Atmospheric Chemistry and Physics, 2016_

## Referee Comment (RC1) · Anonymous Referee #1 · 26 Oct 2016

General comments

This paper is based on 22 years of continuous observations of stratospheric aerosols from two independent ground-based lidar systems located at OHP. This very valuable data set is used in synergy with satellite observations from several instruments. After proper scaling, stratospheric aerosols optical depths between 17 and 30 km retrieved from the OHP lidars and from the satellite instruments are in remarkable agreement. The synergetic analysis of OHP and satellite data allows identifying two "quiescent" time periods during which stratospheric aerosols are "background aerosols" with no contribution from volcanic eruptions. The paper ends with the analysis of the variability (annual cycle) and long term change of the background aerosols. The content of this

manuscript is good, but minor revisions are recommended before publication. The discussion is sometimes difficult to follow and a little too dense, unless the reader is very familiar with the topic. In my opinion, explanations and details are sometimes missing, and equations could be added in some places for more clarity.

Specific comments

Abstract: line 32: define post-Pinatubo era or recall date of eruption

Section 1 Lines 76-80: what are "minor" eruptions in this context? How do they compare with the "strong eruptions" (line 63) of VEI >= 4?

Section 2.1 : OHP lidar instrument:

Lines 113-116: why mention the separate telescope and detection channel if not used for this study? Note that the Hoareau et al. (2013) reference is missing.

Lines 119-121: "the primary low gain channel of LTA. . ...0.03 m2 telescope": are you here describing the "primary, more powerful channel detection channel of LTA" used in this study? The collecting area seems very small. Please clarify.

It is stated that the study is restricted to altitudes above 12-13 km due to saturation issues (lines 115-116). It seems inconsistent with lines 242-244 where it is written that LTA may be affected by incomplete desaturation below 17 km. Please explain the saturation and desaturation issues. Photon counting? Other?

Section 2.2: should the title be "OHP lidars aerosol retrieval"?

Lines 130-133: Please clarify the use of Kb and/or Ke for OHP LiO3S retrievals. Equations would be very helpful, and would clarify the presentation and discussions in Section 2.3 and Section 3.

Lines 141-148: An important source of uncertainty is indeed the actual aerosol loading in the 30-33 km reference region. This is mentioned later (lines 233-234), but could be discussed in more details here. Could CALIOP V4.00 measurements at 30-33 km be

used to estimate OHP calibration biases?

Lines 149-156: this discussion is important and could/should be developed. For instance, the sensitivity to the assumed lidar ratio (line 154) should depend on the optical depth?

Section 2.3 Satellite aerosol sounders

It could be interesting to comment on the fact that some instruments measure a backscatter further converted to extinction whereas other measure directly an extinction. What is the rationale for the selection of the various values of Ke? No conversion to 532 nm is mentioned for GOMOS retrievals. Are they directly at 532 nm? Adding a table summarizing all the conversions applied to the various data sets could be useful.

Lines 184-186: I think that it is important to explain in the text that in Level 1 V4.00 CALIOP products, the nighttime 532 nm channel is calibrated between 36 and 39 km using GMAO Met data. Please explain how the (nonattenuated) backscatter profiles are obtained from CALIOP V4.00 Level 1 data. Because CALIOP is a lidar instrument operating at the same wavelength as LTA, it could make sense to have a dedicated section for CALIOP.

Section 3

Line 200: "zonally averaged"...I suggest to say here " over a 10° latitude belt centered at OHP latitude" (line 217).

Line 204: Can you clarify whether the differences reported in Table 1 are computed for different time periods, depending on the availability of the 2 instruments that are being compared? If yes, these time periods could be specified. I could not find the definition of the differences reported in the table (column – row or the opposite?).

Line 229: "quiescent periods": please explain, because the careful selection of the quiescent periods is presented later in the paper.

Lines 228-234: I don't see a negative bias of the OHP lidars with respect to all satellites above 25 km in Figure 2. Indeed, it seems that OSIRIS has the smallest values. Do you agree? What is the relative difference between the OHP lidars and CALIOP above 25 km? Would it be consistent with the presence of aerosols at 30-33 km, which could be further estimated from CALIOP? Could the difference be due in part to the use of different models (NCEP vs model used in CALIOP algorithm)?

Lines 236-237: I am not totally convinced by the discussion about the lidar ratio, because this issue should not introduce differences between LTA and CALIOP. The notion of desaturation is difficult to understand without specific knowledge of the instruments. I suggest describing these instrument-related difficulties in Section 2.1.

Section 4

Lines 251-253: Table 2 should be introduced in the text. Do you mean VEI=4 or VEI≥4?

Lines 308-314: What are we learning from the fact that no plume from the Calbuco volcano (41°S) is detected at OHP? Why mention this volcano specifically?

Lines 316-333: are the volcanic plumes detected using global satellite coverage or by using the two conditions listed in lines 329-332? My understanding is that global satellite measurements are used to trace the origin of aerosol enhancements detected using the two conditions back to volcanic eruptions. Is it correct? Fig. 6 is complicated and described only in section 5, and introduced before Fig. 5.

Lines 355-356: would it be correct to say that that OHP lidars detect an increase in optical depth end 2010/early 2011, which is traced back to the Merapi volcano using CALIOP observations?

Overall, I think that Section 4 is not very clear and should be reorganized. It is noted that it starts with a long introduction from lines 248 to 314, followed by only one subsection (4.1).

Section 5:

Line 443: please describe Fig 7c.

---

## Referee Comment (RC2) · Anonymous Referee #2 · 28 Oct 2016

This manuscript presents and discusses long-term (past 20 years) measurements of stratospheric aerosols, as measured by two co-located ground-based NDACC lidars (Haute-Provence, France), and by several satellites, including the lidar in-space CALIOP.

After separating the sAOD time-series into two volcanic "quiescent" periods and one volcanic-active period, the take-home message is that the second quiescent period shows a background aerosol level higher than that of the first period, leading to the hypothesis that the enhancement is due to the enhanced transport of aerosols within the Monsoon circulation (ATAL AOD). The authors are wise enough however to recognize that the second quiescent period is very short (2013-2015), and that more measurements in the years to come are necessary before the present conclusions can be firmly confirmed.

The manuscript is well-organized and well-written. Some minor corrections in English syntax are needed, which can be easily managed by one of the native English-speaking co-authors. The discussion occasionally lacks fluidity, for example section 5.2, and the last 3 paragraphs of the discussion/summary. Nevertheless, I recommend publication after some minor corrections, listed below.

Title/Abstract: Since this article is submitted to the NDACC 25th anniversary Special Issue, it would be suitable to explicitly cite NDACC either in the title, or in the abstract (when presenting the OHP lidars).

Section 2.3: Are all extinction coefficient conversions factors (ke) using the same reference of Jäger and Deshler (2002; 2003)? Please specify if so.

Line 200 and Fig. 1: The number of CALIOP samples is much larger than the number of samples from the other instruments. Why not restricting the CALIOP coincidence window within the OHP region instead of the full zonal-mean? This would (maybe) improve the agreement with the lidars, for example during the periods 2007-2008 and 2010-2011.

Fig. 1: Can the aspect ratio of this figure be less elongated (more square) in order to distinguish the various measurements from each other?

Lines 285-287: Fig. 3 does not really show a difference in the e-folding rate for Sarychev and Nabro, it looks more like the background level after Nabro (late 2012) is simply higher than that after Sarychev (early 2010). The authors refer to time-series from CALIOP and OSIRIS . Can they refer to specific publications?

Lines 329-330 and fig.3: How do the monthly-mean sAOD uncertainties compare with the 1-sigma threshold level set to determine what is quiescent and what is not? Although there is a risk to overload Fig. 3, it would be interesting to overplot vertical

bars to denote +/- 1 standard uncertainty for each monthly mean sAOD sample plotted. In any case, some text should be added to discuss the relative magnitude of this uncertainty and the 1-sigma threshold value. This will determine whether the observed increase can be considered statistically significant or not.

Lines 440-441: See above comment on line 200: For CALIOP, it would make more sense to use an average over a longitude band centered over OHP (e.g., +/-20 deg) rather than a full zonal mean.

Last Paragraph (Lines 558-563): The authors should also emphasize on the critical need for ground-based lidar observations in the next decade, as there will possibly be a gap in aerosol profilers from space after CALIOP has ceased operation.

––––––––––––––––––––––––––––––

---

## Short Comment (SC1) · 7 Nov 2016

I have a question regarding Figure 4. The plot surprised me because it did not show Sarychev Peak aerosols in the full altitude range of the eruption's impact. See Jegou et al. (http://www.atmos-chem-phys.net/13/6533/2013/) and their Figure 6. See a Sarychev layer at ∼22 km in mid-July. Jegou et al. did not bring attention to this feature, but it is real. The OHP NDACC Rayleigh lidar data for July 2009 reveal a strong aerosol peak at ∼22 km on 16 July. This profile does not show up in Figure 4. My question is why did this layer show up in Jegou's OHP analysis and not in the current paper? How consistent is the NDACC OHP lidar archive with respect to the data presented in this paper?

It seems to me that your paper has an opportunity to draw attention to this previously unreported aspect of the Sarychev eruption: injection heights exceeding 20 km. The 20+ km aerosol is fully evident in CALIPSO data as well. Expedited imagery shows the high aerosols over France during this time frame. Here are 2 CALIPSO slices from the time of the OHP sighting: https://www-calipso.larc.nasa.gov/products/lidar/browse_images/show_detail.php?s=production&v=V3-01&browse_date=2009-07-16&orbit_time=01-35-53&page=1&granule_name=CAL_LID_L1-ValStage1-V3-01.2009-07-16T01-35-53ZN.hdf https://www-calipso.larc.nasa.gov/products/lidar/browse_images/show_detail.php?s=production&v=V3-01&browse_date=2009-07-17&orbit_time=02-19-06&page=1&granule_name=CAL_LID_L1-ValStage1-V3-01.2009-07-17T02-19-06ZN.hdf Here's a shot of the high aerosol a couple weeks earlier, over Asia, before it got to Europe. https://www-calipso.larc.nasa.gov/products/lidar/browse_images/show_detail.php?s=production&v=V3-01&browse_date=2009-06-30&orbit_time=21-23-20&page=1&granule_name=CAL_LID_L1-ValStage1-V3-01.2009-06-30T21-23-20ZN.hdf
* * *

---

## Referee Comment (RC3) · Anonymous Referee #3 · 9 Nov 2016

Lidar aerosol observations from Observatoire de Haute-Provence (OHP) in Southern France have been made since the 1980s, but have not been summarized in the literature since the early 1990s, which is the purpose of this paper. After reviewing the basis of the record and the conversion to a common 532 nm wavelength, the authors convert lidar backscatter measurements to extinction to calculate stratospheric aerosol optical depth (sAOD) from 17-30 km for comparison with a number of satellite aerosol instruments. The agreement is remarkably good and this result will be a nice addition to the literature.

The authors then use the global CALIOP data base to begin making sweeping conclusions about the processes controlling the stratospheric aerosol observed at OHP.

While the measurements presented are at times consistent with the statements about processes, the statements made about processes are not definitively established with the analysis shown. Thus the conclusions extend beyond what is established in this paper and major sections of the paper should be reconsidered and carefully rewritten to reflect that the processes described are not definitive and may be tempered by many complicating factors.

When the authors stay close to their data and make a definitive separation between measurements classified as volcanic and non-volcanic, and present data representative of these two states, then useful results are obtained. This should be more carefully presented and discussed, and the discussions of global processes treated less definitively. Here are some details.

265-267. The authors state, "Both SAGE II and OHP lidars report an average background sAOD1730 for the "reference" quiescent period of $2.3×10-3 \pm 2.4\%$ (2 SE), which is marked in Fig. 3 by dashed line and grey shading, indicating 1-$\sigma$ range of values." This statement implies that sAOD should be between 0.00225 and 0.00235 since 2.4% of 0.0023 is 0.00005, yet the range shown in the figure is much larger than this. The authors claim that 2.4% is 2 SE, which, the reader is left to assume, means 2 standard errors. Then the authors say the shading represents 1-$\sigma$, without explanation. So in the end the reader is unsure what is shown in the figure, but it seems to be larger than 2.4% of the mean value quoted and how does 1-$\sigma$ compare with 2 SE?

231-234. It is very difficult for the reader to understand how the figure supports these statements. Above 25 km the lidar data do not show any particularly different bias compared to satellite than below in the left panel of Fig. 2. The lidar data lie within the symbols for both SAGE and OSIRIS. On the right panel the lidar data split the satellite data and the agreement is overall better than below 25 km. Below 25 km the agreement with CALIOP remains good but is worse OSIRIS and OMPS.

288-301. Surely the differences between the plumes of Sarychev and Nabro are primarily driven by the significantly different latitudes of the two eruptions, compared to the latitude of OHP, and the dominance of the mixing by zonal flow in the stratosphere. Sarychev, at nearly the same latitude as OHP, is detected very early and the volcanic plume appears as pulses of aerosol, as these pulses are advected around the Earth before they are significantly mixed by the general flow. In contrast the aerosol from Nabro is already well mixed by the general flow prior to its arrival at OHP, 45 days after the eruption. To effectively compare the evolution of these two eruptions the color scales should be adjusted to both start at the day of first detection of Nabro, 45 days after each eruption. All profiles prior to this time from Sarychev could be indicated as black profiles.

319-320. "The plumes of more distant (tropical) eruptions are not always obvious in sAOD series." What is a more distant tropical eruption? Nabro is tropical. Considering the dominant zonal flow does the longitude of a tropical eruption make a big difference? Why are these "more distant tropical eruptions" not evident in sAOD series? Is this sAOD now meant to only imply sAOD at OHP? Distant and close tropical eruptions will make a difference in sAOD depending on where sAOD is measured, but the reader is left to guess what is intended. The text implies that the plume from a volcanic eruption has a rather direct stratospheric transport to the mid latitudes from a tropical eruption, but doesn't the dominant zonal flow in the extra tropical stratosphere confound this idea?

336-337 and Fig. 5. "Aged" is not a very descriptive term. Better would be some consistency such that the volcanic curves represent an average of the measurements over some specified time period, which ideally would be the same time after each eruption.

353-357. The CALIOP data are far from clearly supporting the suggestion that the plume from Merapi was observed at OHP. The structure in the CALIOP data at OHP latitude in early 2011 which coincides with the blue shading in the OHP data has an origin prior to Merapi, whereas it is not obvious that the plume from Merapi is still intact at

45°N. The sAOD1519 from CALIOP is 2e-3 to 3e-3 compared to 5e-3 at OHP. In contrast after Nabro in mid to late 2011 the CALIOP data display a significant increase in aerosol at OHP latitudes whereas OHP sAOD is hardly larger than the value attributed to Merapi. Such discrepancies raise questions about how well these two data sets really agree, particularly at these altitudes. Is this reflective of the differences between the OHP and CALIOP measurements below 16 km in Fig. 2b. This seems unlikely. Figure 6, in the discrepancies of the timing between OHP sAOD and CALIOP sAOD for both Nabro and Sarychev, raises question about the correspondence of these two data sets. At the very least the timing of Sarychev, Nabro, and many of the aerosol minima appear to be displaced, with OHP lidars lagging the CALIOP data.

367-372. Fig. 6 displays 10 years of CALIOP AOD from 15-19 km from 60S to 60 N. What fraction of the troposphere is included here? Certainly in the equatorial and tropical regions there is about 1-2 km of tropospheric data since the tropopause is typically near 17 km. The upper troposphere can be quite clean if there is deep convection or it can be influenced by tropospheric aerosol. To attribute all the data shown in Fig. 6 to the stratosphere is misleading. Here the authors want to suggest based on signatures, clouded by the uncertainties just mentioned, that 4 of these 10 years display evidence of the ATAL. But how would the ATAL be separated from other aerosol laden air from the upper troposphere? What other evidence is there to link this slight change in AOD to the ATAL? Is it really so clear in terms of the timing of these events? How similar is it? Finally this is a paper about the OHP lidar record not a broad scale interpretation of the CALIOP data from 60 S to 60 N. If the latter is the intent then do a complete job on the CALIOP observations. Here the intent appears to be on the OHP lidars. If so then there should be a better discussion of when the CALIOP is in agreement with OHP, when it is not, and why there are differences.

373-382. This picture is a bit less clear than suggested. Many of the Northern Hemisphere low aerosol tongues are rather discontinuous even when volcanoes are not involved. The lidar and CALIOP timing of the low aerosol load are different. While

there is some evidence for the author's assertion, it is far from definitive, and other processes may be involved. The influence of the troposphere on the AOD displayed is unclear. It is also not clear to what extent a higher summer tropopause would affect the OHP data compared to a lower tropopause in the winter. If the authors wish to pursue this type of interpretation of the CALIOP data they should consider preparing a paper focused on such analysis of the CALIOP data and not add it as a sidelight to this paper about OHP lidars.

Fig. 7a and 7b display several discrepancies. CALIOP data display the expected Junge layer with minimums below 18 and above 24 km, and a maximum near 20 km throughout the year. OHP suggests a significant modulation of the Junge layer with a decrease of AOD from 1.08 to 1.04 from April to December which is not seen in the CALIOP data. Is this seen in other data sets? It is not clear what would cause this modulation of the Junge layer. The CALIOP data do not show a strong increase in aerosol near 16 km in the autumn. The authors explain this away as due to zonal averaging. But really is the connection so immediate, from the Asian monsoon to 45°N, that the ATAL would only appear in the OHP data? Is the ATAL signal so small that it is diluted with the zonal average, even though that average would incorporate much more of the Asian monsoon outflow than would reach OHP?

Why are the time periods covered by Fig. 7a, 7b so different? Is there a point to be made about similarities of any non-volcanic period, or is the point to show how similar the OHP lidars are to CALIOP? If the latter then wouldn't it be better to compare the same time frames?

525-526. Calling the authors' explanations for the observations "rather robust" is not justified in this reviewer's mind, and suggesting there may be alternate explanations, which are not explored, but should be, is less than genuine at this point in the conclusions.

The discussion section is a recap of the conclusions reached based on the analysis

discussed above which I find incomplete and perhaps misleading. The models the authors have to characterize the data are too simplistic and ignore many complicating factors.

Minor comments:

870. embedded panel? Do the authors mean the legend?

175. I am not quite sure what is meant by occultations for a limb scatter instrument. What is being occluded?

291. 3.4 units? Do the authors mean a scattering ratio of 3.4?

307-308. Why do the satellite measurements not agree with the optical depth decrease after January 2015 observed by the OHP lidars? Rather the satellites remain elevated at the January level.

309. This comment on Calbuco is not really necessary here since it does not affect post Nabro OHP and forces the reader to look ahead to Fig. 6 to verify the statement, which is then called out of order.

323. What is the partial sAOD examined? Is it the same for all satellites? It should be stated what the AOD covers.

324. Another call out to Fig. 6 out of order. Should the figure orders be reversed?

329-332. "monthly-mean sAOD1730 and SR" where? Is this for OHP only or does it include all the satellite data? In ii) specify the "reference" quiescent period, e.g. 1997-2003. 336. Concerning the quiescent period, the text and Fig. 5 caption state 1997-2003, the legend in the figure states 1998-2003? These should be consistent. 365-366. "The enhanced poleward transport into the winter hemisphere is exhibited by meridional wind vectors in Fig. 6." Then according to the figure there is no meridional wind after 2009. Is this correct?

Fig. 7 caption. The reader does not know what is meant by "SR from OHP LiO3S lidar

for selected volcanically-quiescent periods . . ." What is the selection based on? Is it all non-volcanic periods or just select periods?

428-429."Importantly, for any quiescent subperiod over the course of 22 yr OHP series, the pattern is essentially the same." The OHP lidar record is only 22 years long, so what does this statement mean? Do the authors mean any quiescent subperiod within the 22 year data record?

––––––––––––––––––––

---

## Author Comment (AC1) · 4 Jan 2017

The OHP lidar data presented in the ACPD version of the article are restricted to the altitude range between 15 and 31 km. The lower boundary (15 km) is justified in the article in consideration of the signal saturation issues (recalling that the lidars are optimized for the stratosphere) and the presence of cirrus clouds above OHP up to 14 km. In Sect.3 (Intercomparison of OHP lidars and satellite sounders) it was pointed out that the aerosol extinction profiles obtained by the lidars are high-biased with respect to the satellite profiles below 17 km and the bias increases with decreasing altitude (Fig. 2). The same inference could be made on the basis of Fig. 6 and 7, where the lidar data are qualitatively compared with those of CALIOP. Several questions were raised

by the referees, which regarded the OHP lidar data quality in the lowermost strato-sphere (LMS) and the reasons for the overestimation of aerosol backscatter/extinction below 17 km by OHP lidar (as could be inferred from the comparison with CALIOP). Referee #1 noted some inconsistency in the mention of altitude range limitation for the lidar aerosol retrieval and questioned the proposed explanation of the LMS discrepan-cies. Referee #3 expressed concerns on the consistency between OHP and CALIOP data presented in Fig. 6. In addition, a short comment posted by M. Fromm raised a question regarding the absence of signatures of the Sarychev plume above 20 km in an ensemble of vertical profiles reported in Fig. 4. The received remarks led us to revisit the OHP lidar data and investigate the issues with the data at lower altitudes. It was found that a large fraction of the aerosol profiles were affected by overcorrection for the signal saturation, which resulted in a positive bias below about 17 km, increas-ing towards the tropopause level. Figure AC1.1 shows how the overcorrection for the signal saturation affects the retrieval of a single scattering ratio profile. The red dashed ("overcorrected") profile, while being identical to the correctly retrieved profile above 20 km, exhibits a significant positive bias in the LMS. The incorrect retrieval had a minor effect on the integrated aerosol optical depth above 17 km (sAOD1730), a fair effect on the sAOD1519 series and a major effect on the data below 15 km. Thus, all the lidar data had to be fully reprocessed using the correct parameters applied to signal treatment, which yielded reasonable data down to the tropopause. The revised LiO3S AOD series differ from the initial ones on average by +1.9% (sAOD1730), −16.9% (sAOD1519) and −34.6% for the AOD between the tropopause and 15 km. It should be clarified here that while the study makes use of two different OHP lidars (LiO3S and LTA) only the data of LiO3S could be recovered down to the tropopause. The LTA system, being optimized for the middle atmosphere, could not be used to obtain use-ful information on the aerosol in the LMS. In addition to the improved signal treatment for both lidars, the resulting data have been subjected to a manual profile-by-profile screening and filtering of cirrus clouds occurring around the tropopause. A particular attention was given to the periods of volcanic plumes sampling, namely the detection

of Okmok, Sarychev and Nabro plumes. This effort showed that a semi-automated screening procedure applied to the initial version of the data has left behind some of the useful measurements, e.g. a strong aerosol peak at 21.5 km originating from the Sarychev eruption or an early detection of Nabro plume (15 days after the eruption as opposed to 45 days reported initially). All the analysis involving OHP lidar observations as well as the respective figures and tables have been updated using the reprocessed data set. The lower boundary of all the plots that show aerosol vertical distribution have been extended below 15 km. While the scientific interpretation of the results remains intact, the reprocessed OHP data appear more consistent with those of CALIOP in the LMS and the updated figures provide a better insight into the annual cycle of aerosol in the lowermost stratosphere and its long-term change. The reprocessed data induced a number of changes to the figures, tables and text. Most of them are mentioned in the replies to reviewers. We provide below the list of the most important changes that were made beside the revision suggested directly by the referees.

Changes to figures. Fig. 1. The lidar monthly-mean series appears somewhat less noisy Fig. 2a.b. Lower altitude limit extended down to 12 km, LiO3s extinction profiles no longer exhibits a positive bias compared to satellites. LTA extinction profile is re-moved from the figure. Fig. 3. Mean OHP lidars sAOD1730 appears less noisy Fig. 4a,b. Lower altitude limit extended down to 12 km, dates of first detection corrected, color scale revised, new (previously discarded) profiles included. Fig. 5. Lower altitude limit extended down to 13 km, average SR profile for the "reference" period (solid black) no longer shows positive bending in the LMS Fig. 7a,b Lower altitude limit extended down to 13.5 km, color scale adjusted to better demonstrate the annual pattern. Fig. 7c. Lower altitude limit extended down to 13.5 km Fig. 8a. In addition to some minor alteration of the month-altitude pattern of the AOD change related to the new repro-cessing, the updated plot is fixed in terms of the vertical scaling, which was found to be wrong for the initial version of the plot.

Principal changes to the text Section 3 (Intercomparison of OHP lidars and satellites

sounders). Paragraphs 1-2. All the figures obtained on the basis of OHP lidar data have been updated Last paragraph. The discussion around Fig. 2 showing intercomparison of extinction profiles has been fully revised. Section 4 (Volcanic plumes and quiescent periods) Discussion around Fig. 4 showing the detection of Sarychev and Nabro plumes (Sect. 4.2.1) has been entirely revised. The period posterior to Merapi eruption is excluded from the list of volcanically-perturbed periods for OHP. The mention of it in Sect. 4.4 (last paragraph in former Sect. 4.1) has been removed. Section 5.1 (Annual cycle) Paragraphs 3-5 discussing Fig. 7a and 7b have been revised following the changes in Fig. 7.

Changes to tables. Table 2 (former Table 1). The intercomparison figures involving OHP data have been updated: the new values of relative difference and correlation coefficients indicate slightly improved agreement. Table 3 (former Table 2). The period posterior to Merapi eruption is excluded from the list of volcanically-perturbed periods for OHP.

Please also note the supplement to this comment:
http://www.atmos-chem-phys-discuss.net/acp-2016-846/acp-2016-846-AC1-supplement.pdf

**OHP LiO3S 2015 9 22**

- - - "overcorrected" profile
—— correct retrieval

*Z* (km)

SR@355 nm

**Fig. 1.** Figure AC1.1 Vertical profiles of scattering ratio @ 355 nm from OHP LiO3S lidar showing the effect of overcorrection for signal saturation on the aerosol retrieval in the lower stratosphere.

---

## Author Comment (AC2) · 4 Jan 2017

We express our gratitude to Anonymous Referee #1 for constructive remarks on the manuscript. Please check the pdf file in supplement, which contains the formatted text for easier reading.

Abstract: line 32: define post-Pinatubo era or recall date of eruption The respective sentence was modified: "...during the last two decades"

Section 1 Lines 76-80: what are "minor" eruptions in this context? How do they compare with the "strong eruptions" (line 63) of VEI >= 4? Compared to VEI≥5 eruptions, the VEI=4 ones can be termed "minor" considering their impact on stratospheric

aerosol load. However, it would be more correct to term the VEI=4 eruptions "moderate" The sentence was modified: "...the increase was primarily caused by moderate volcanic eruptions with VEI=4, whose impact..."

Lines 113-116: why mention the separate telescope and detection channel if not used for this study? Note that the Hoareau et al. (2013) reference is missing. This information is addressed to the users of NDACC aerosol data who may already be familiar with OHP aerosol data and should be aware that the present study makes use of a different measurement source. A reference to Hoareau et al. (2013) was added.

Lines 119-121: "the primary low gain channel of LTA. . ...0.03 m2 telescope": are you here describing the "primary, more powerful channel detection channel of LTA" used in this study? The collecting area seems very small. Please clarify. The LTA system features several telescopes and corresponding detection channels, including Rayleigh/Mie and Raman channels. The Rayleigh/Mie channels A (0.88 m2 telescope) and B (0.03 m2 telescope) represent the high and low gain channels of the temperature lidar. The cirrus/aerosol channel E uses a separate telescope of 0.03 m2 that is the same size as for channel B. It is thus incorrect to refer to channel B as "more powerful". However, the advantage of using channel B instead of the cirrus/aerosol channel E is that the former was more regularly maintained and has less temporal gaps in the measurements. In addition, the electronic range-gating of the photomultiplier in the B channel is set to 12 km altitude (as opposed to 5 km for channel E), which improves the signal-to-noise ratio at the calibration levels, whilst limiting the useful measurement range to altitudes above 14 km. The collecting area of the LTA low-gain B channel is not large, however this is compensated by a powerful laser (17 W, upgraded to 24 W after 2013). This translates to a total lidar power of 0.54 (0.74) W•m2, which is larger than some of the NDACC aerosol lidars and comparable to CALIOP (1.73 W•m2). The respective paragraph was modified as follows: "The LTA system includes a separate telescope and detection channel for clouds and aerosol (Chazette et al., 1995; Keckhut et al., 2005; Hoareau et al., 2013). In contrast to the pervious studies we use for the first

time the primary low-gain detection channel of LTA system for stratospheric aerosol retrieval. This choice benefits from lesser measurement gaps thanks to a more regular maintenance and better signal-to-noise ratio of the LTA low-gain channel, which is achieved thanks to the electronic range-gating adjusted to 12 km altitude. This configuration reduces the signal-induced noise at mid-stratospheric levels whilst limiting the useful measurement range to altitudes above 14 km."

It is stated that the study is restricted to altitudes above 12-13 km due to saturation issues (lines 115-116). It seems inconsistent with lines 242-244 where it is written that LTA may be affected by incomplete desaturation below 17 km. Please explain the saturation and desaturation issues. Photon counting? Other? The last paragraph of the Section 3 is no longer valid (see. AC1 "Reprocessing of the OHP lidar data and related changes to the manuscript") and had to be revised completely. We no longer restrict to the altitudes above 15 km when using LiO3S lidar measurements since the data were reprocessed using a proper correction for signal saturation affecting the aerosol retrieval at lower levels. However, this exercise could not be applied to LTA data due to range-gating limiting the useful measurement range to altitude above 14 km. The last paragraph of Sect. 3 is updated as follows: "Figure 2 displays a comparison of aerosol extinction profiles averaged over two 20-month volcanically-quiescent periods 2002-2003 and 2013-2014 covered by time-overlapping observations by two different triplets of satellite sounders. The comparison reveals close agreement between OHP lidar, SAGE II, GOMOS and OSIRIS (Fig. 2a) above 15 km and somewhat poorer agreement below. Fig. 2b suggests a good agreement between OHP lidar and CALIOP (relative difference 5-10%) throughout the entire range of altitudes except the uppermost layer above 25 km, where OHP lidar is 15-20 % low with respect to CALIOP. This feature may be related to an error in lidar calibration, relying on the assumption of the absence of aerosol above 30 km, which – as suggested by CALIOP data calibrated at higher altitudes - may not always be the case. The other two satellite sounders covering 2013-2014 period – OSIRIS and OMPS - show somewhat larger discrepancies (reaching 30% ) with OHP lidar and CALIOP in the uppermost and lowermost layers.

This discrepancy may be due to the use of a fixed lidar ratio and wavelength exponents, which may vary with height depending on the size distribution of aerosol."

Section 2.2: should the title be "OHP lidars aerosol retrieval"? The title was modified as advised.

Lines 130-133: Please clarify the use of Kb and/or Ke for OHP LiO3S retrievals. Equations would be very helpful, and would clarify the presentation and discussions in Section 2.3 and Section 3. Four equations defining the scattering ratio and the wavelength conversion for backscatter, extinction and scattering ratio have been added. The text was modified accordingly.

Lines 141-148: An important source of uncertainty is indeed the actual aerosol loading in the 30-33 km reference region. This is mentioned later (lines 233-234), but could be discussed in more details here. Could CALIOP V4.00 measurements at 30-33 km be used to estimate OHP calibration biases? The assumption on the zero aerosol loading above 30 km (at least in the absence of major eruptions of VEI≥5) is a commonly used approach for aerosol retrieval using ground-based lidars. In principle, CALIOP measurements above 30 km could be used to recalibrate the ground-based lidar retrieval, however we prefer to follow the traditional approach in the context of this study, in which the ground-based data are obtained in an independent way and are then used to identify the discrepancies with satellite data and to discuss their possible sources. As follows from the comparison in Sect. 3, the effect of the possible calibration bias is limited to altitudes above 25 km and the associated bias in the extinction profile does not exceed 15%. For the integrated values of extinction (sAOD) the calibration bias has a negligible effect.

Lines 149-156: this discussion is important and could/should be developed. For instance, the sensitivity to the assumed lidar ratio (line 154) should depend on the optical depth? Indeed, the sensitivity of scattering ratio and backscatter coefficient depend on the optical depth. However, even under high aerosol load conditions (e.g. peak

of Nabro signal in sAOD), this sensitivity is still very low. The text was modified as follows: "We note that the uncertainty in the assumed lidar ratio has a limited effect on the derived values of backscatter coefficient and scattering ratio. For example, the sensitivity of the stratospheric mean $\beta$aero to the assumed lidar ratio was estimated at ~0.15 %/sr under background aerosol conditions (September 2005) and ~0.23 %/sr under volcanically-perturbed conditions (September 2011). Our estimates are compatible with those provided by Sakai et al. (2016). It should be noted that the error in lidar ratio has a larger effect on aerosol extinction and optical depth, whose uncertainty may thus be somewhat larger."

Section 2.3 Satellite aerosol sounders It could be interesting to comment on the fact that some instruments measure a backscatter further converted to extinction whereas other measure directly an extinction. What is the rationale for the selection of the various values of Ke? No conversion to 532 nm is mentioned for GOMOS retrievals. Are they directly at 532 nm? Adding a table summarizing all the conversions applied to the various data sets could be useful. The value of wavelength (Angstrom) exponent depends on the wavelength pair. For the 355 nm -532 nm pair ke value was adapted from Jager and Deshler (2002; 2003). The same value (ke=−1.6) was used for SAGE II (525 nm) and GOMOS (550 nm), for which the wavelength conversion means multiplying the extinction by a factor of 0.979 and 1.055 respectively. For OSIRIS (750 nm) and OMPS (675 nm) the wavelength exponents were suggested by the instrument PIs. The following paragraph was added in the beginning of Sect. 2.3: "Over the course of the last two decades stratospheric aerosol observations from space were conducted by various satellite missions, exploiting different measurement techniques: solar and stellar occultation, limb scattering as well as nadir-viewing lidar. We use five satellite-based datasets, altogether covering the time span of OHP lidar observations. " The following paragraph was added in the end of Sect. 2.3: "It should be noted that among the passive satellite sounders SAGE II and GOMOS measure aerosol extinction, whereas OSIRIS and OMPS measure limb-scattered radiation, from which aerosol extinction is then retrieved. In contrast, CALIOP instrument, based on active sounding

technique, measures aerosol backscatter. In order to compare OHP lidars and satellite instruments all data sets were converted to extinction at a common wavelength of 532 nm. Table 1 summarizes the wavelength exponents $\kappa$e used for conversion (eq. 3) and the time spans of data sets involved in the present analysis." A new table was added: "Table 1. Stratospheric aerosol sensors exploited: (columns, left to right) name of instrument, operating wavelength, wavelength exponent for extinction $\kappa$e used for conversion to 532 nm, conversion factor (see eq. 3), time span of available data."

Lines 184-186: I think that it is important to explain in the text that in Level 1 V4.00 CALIOP products, the nighttime 532 nm channel is calibrated between 36 and 39 km using GMAO Met data. Please explain how the (nonattenuated) backscatter profiles are obtained from CALIOP V4.00 Level 1 data. Because CALIOP is a lidar instrument operating at the same wavelength as LTA, it could make sense to have a dedicated section for CALIOP. Since the paper is focused on the observations by OHP lidars whereas the information on CALIOP and stratospheric aerosol retrieval is readily available in various articles by J.-P. Vernier et al. cited throughout this manuscript, we prefer to avoid dedicating a separate section to CALIOP. However the respective paragraph was updated with more details as follows: "CALIOP (Cloud-Aerosol Lidar with Orthogonal Polarization) onboard CALIPSO satellite platform is a nadir-viewing active sounder (Winker et al., 2010). Operational since June 2006, CALIOP provides range-resolved measurements of elastic backscatter at 532 nm and 1064 nm with a vertical resolution of around 200 m in the stratosphere. CALIOP lidar makes use of a Nd:Yag laser operating at 20.2 Hz with a 110 mJ/pulse power and a 0.78 m2 telescope. The data used here are based on night-time 532 nm level 1B version 4.00 product, post-processed using a treatment described by Vernier et al. (2009). The total attenuated backscatter profiles from CALIOP are corrected for molecular attenuation and ozone absorption after adjusting the calibration altitude to 36-39 km. The attenuation by aerosol, constituting less than 1% at 15 km during background aerosol conditions, is neglected. Data below clouds are removed from the analysis. The scattering ratio profiles are obtained using molecular backscatter computed using NASA Global Modeling and Assimilation Office

(GMAO) data. The backscatter data of CALIOP are cloud-cleared in the upper tropo-sphere using a depolarization ratio threshold of 5%. The conversion of backscatter to extinction is done using lidar ratio of 50 sr."

Section 3 Line 200: "zonally averaged"...I suggest to say here " over a 10ậŮę latitude belt centered at OHP latitude" (line 217). The respective sentence has been modified: "...whereas the satellite values (monthly- and zonally-averaged over a 10° latitude belt centered at OHP latitude) contain..."

Line 204: Can you clarify whether the differences reported in Table 1 are computed for different time periods, depending on the availability of the 2 instruments that are being compared? If yes, these time periods could be specified. I could not find the definition of the differences reported in the table (column – row or the opposite?). The following sentence has been added at the end of the first paragraph of Sect. 3: "Note that the differences reported are computed for different time periods, depending on the avail-ability of the data of each instrument ." The caption of Tab. 2 (former Tab.1) is updated as follows: Table 2. Intercomparison of stratospheric Aerosol Optical Depth between 17 and 30 km (sAOD1730) series displayed in Fig. 1. Mean relative difference ∆mean ± 2 standard errors (top) and correlation coefficient R (bottom). Relative difference in the top panel is calculated as where X is the sAOD1730 value averaged over the en-tire observation time span of the respective instrument (see Tab. 1) or the mean of all satellite instruments (last column). Please note that the values in Tab. 2, computation of which involved OHP data, have been updated after OHP data reprocessing (see. AC1 "Reprocessing of the OHP lidar data and related changes to the manuscript"). The associated text in the beginning of Sect. 3 has been updated accordingly.

Line 229: "quiescent periods": please explain, because the careful selection of the qui-escent periods is presented later in the paper. The beginning of last paragraph in Sect. 3 has been modified as follows: "Figure 2 displays a comparison of aerosol extinc-tion profiles averaged over two 20-month periods 2002-2003 and 2013-2014 covered by time-overlapping observations by two different triplets of satellite sounders. These

periods are also characterized by a stable aerosol load that is without strong enhancements due to volcanic eruption."

Lines 228-234: I don't see a negative bias of the OHP lidars with respect to all satellites above 25 km in Figure 2. Indeed, it seems that OSIRIS has the smallest values. Do you agree? What is the relative difference between the OHP lidars and CALIOP above 25 km? Would it be consistent with the presence of aerosols at 30-33 km, which could be further estimated from CALIOP? Could the difference be due in part to the use of different models (NCEP vs model used in CALIOP algorithm)? Please note that Figure 2 and respective discussion provided in Sect. 3 have been fully revised following the result of OHP data reprocessing. There is some negative bias of OHP LiO3S lidar compared to CALIOP above 25 km with relative difference of 15-20 %. It is hardly discernible in the plot as the extinction is very small at these altitudes. This difference may potentially be due in some part to the use of different models (NCEP and GMAO), however it is more likely that the negative bias of the OHP profile is due to higher altitude of calibration in CALIOP data retrieval. Nevertheless, given very small aerosol extinction above the Junge layer, the calibration issue has a negligible effect on the AOD values as can be concluded from the intercomparison. The discussion around Fig. 2 has been revised as follows: "OHP lidar and CALIOP capture well and agree on the main features of background aerosol annual cycle in the lower mid-stratosphere, whereas above 25 km CALIOP shows higher SR values compared to OHP lidar and somewhat less pronounced annual cycle. This may be due to higher altitude of calibration for CALIOP retrieval and the use of different atmospheric models for deriving molecular backscatter (Sect. 2.3 and 3). "

The following text was added in Sect. 5: "OHP lidar and CALIOP capture well and agree on the main features of background aerosol annual cycle in the lower mid-stratosphere, whereas above 25 km CALIOP shows higher SR values compared to OHP lidar and somewhat less pronounced annual cycle. This may be due to higher altitude of calibration for CALIOP retrieval and the use of different atmospheric models

for deriving molecular backscatter (Sect. 2.3 and 3)."

Lines 236-237: I am not totally convinced by the discussion about the lidar ratio, because this issue should not introduce differences between LTA and CALIOP. The notion of desaturation is difficult to understand without specific knowledge of the instruments. I suggest describing these instrument-related difficulties in Section 2.1. Indeed, the lidar ratio has nothing to do with the positive bias in OHP lidar profiles at lower levels as was concluded in the initial version of the manuscript, i.e. before the OHP data reprocessing. The discussion of altitude limitation for LTA lidar are now restricted to Sect. 2.1. Figure 2 and respective discussion provided in Sect. 3 have been fully revised following the result of OHP data reprocessing. The plots in Fig. 2 were extended to 12 km altitude but LTA profile was removed from the plot. The text in the end of Sect. 3 has been rewritten: "Figure 2 displays a comparison of aerosol extinction profiles averaged over two 20-month periods 2002-2003 and 2013-2014 covered by time-overlapping observations by two different triplets of satellite sounders. These periods are also characterized by a stable aerosol load that is without strong enhancements due to volcanic eruption The comparison reveals close agreement between OHP lidar, SAGE II, GOMOS and OSIRIS (Fig. 2a) above 15 km and somewhat poorer agreement below. Fig. 2b suggests a good agreement between OHP lidar and CALIOP (relative difference 5-10%) throughout the entire range of altitudes except the uppermost layer above 25 km, where OHP lidar is 15-20 % low with respect to CALIOP. This feature may be related to an error in lidar calibration, relying on the assumption of the absence of aerosol above 30 km, which – as suggested by CALIOP data calibrated at higher altitudes - may not always be the case. The other two satellite sounders covering 2013-2014 period – OSIRIS and OMPS - show somewhat larger discrepancies with OHP lidar and CALIOP, reaching 30% in the uppermost and lowermost layers. This discrepancy may be due to the use of the fixed wavelength exponents, which may vary with height depending on the size distribution of aerosol."

Section 4 Lines 251-253: Table 2 should be introduced in the text. Do you mean VEI=4

or VEI≥4? Table 3 (former Table 2) is introduced in the end of the first paragraph of Sect. 4: "The selection criteria are described hereinafter (Sect. 4.4), whereas the eruptions and periods affected are summarized in Table 3." We meant VEI=4 eruptions, as there were no VEI>4 eruptions since Pinatubo. Putting "VEI≥4" instead may make a reader wonder if there were eruptions stronger than VEI 4 since 1994.

Lines 308-314: What are we learning from the fact that no plume from the Calbuco volcano (41˚S) is detected at OHP? Why mention this volcano specifically? Indeed, the mention of Calbuco eruption and its possible effect on stratospheric aerosol at Northern mid-latitude is beyond the scope of this paper, based on the data up to October 2015. The respective text has been removed.

Lines 316-333: are the volcanic plumes detected using global satellite coverage or by using the two conditions listed in lines 329-332? My understanding is that global satellite measurements are used to trace the origin of aerosol enhancements detected using the two conditions back to volcanic eruptions. Is it correct? Fig. 6 is complicated and described only in section 5, and introduced before Fig. 5. The detection of mid-latitude volcanic plumes (as well as Nabro) is straightforward as these eruptions left strong signatures in scattering ratio profiles and sAOD series. The detection of volcanic plumes from remote tropical eruptions using solely OHP observations is too ambiguous because the enhancements in sAOD are less pronounced and because the aerosol variability may be caused by processes other than volcanism. Therefore the plumes of tropical eruptions are first detected by visual examination of the satellite time-latitude sections and then, if a plume is found to extend Northward beyond the tropical belt, the two criteria are applied to the lidar data in order to determine the extent of a volcanically-perturbed period. In other words, the satellite data are used to detect a plume, whereas the OHP data are used to determine the duration of the respective volcanic period. The second (former first) paragraph of Sect. 4.4 (former 4.1) was revised as follows: "Volcanic plumes were detected by examining time-latitude sections of sAOD1730 and sAOD1519 from all satellite records (example for CALIOP

is provided hereinafter in Sect. 5). If a plume was found to extend beyond the tropical belt towards the Northern extra-tropics, the OHP lidar monthly-mean sAOD1730 values and SR profiles posterior to the eruption were compared against those averaged over the "reference" quiescent period 1997-2003. This way, the presence of a plume at OHP and the temporal extent of the corresponding volcanic period were determined. In other words, the satellite data are used to detect a plume, whereas OHP lidar data were used to determine the duration of the respective volcanic period at OHP latitude. Thus, a period is considered as volcanically-perturbed if a plume occurs in the Northern hemisphere and if both of the following two conditions are fulfilled in OHP observation posterior to the eruption:" The introduction of Fig. 6 in this section is indeed premature, however it provides a great aid in understanding how the volcanic plumes are detected using satellite data. The sentence has been nevertheless modified : "...(example for CALIOP is provided hereinafter in Sect. 5).

Lines 355-356: would it be correct to say that that OHP lidars detect an increase in optical depth end 2010/early 2011, which is traced back to the Merapi volcano using CALIOP observations? As a matter of fact, it is the other way around. As explained in response to the previous remark, first the plume of Merapi is detected by CALIOP then the OHP profiles are checked to fulfil the criteria. After the OHP data reprocessing, both criteria are no longer fulfilled, thus the period in late 2010/early 2011 is no longer considered as volcanically perturbed. Indeed, as noted by Referee #3, the CALIOP data are far from supporting the suggestion that Merapi plume was transported to OHP latitude. Furthermore, the OSIRIS data suggest that Merapi plume hardly reached beyond 30° N and had an impact mainly in the Southern hemisphere. The last paragraph of Sect. 4.4 (former 4.1) describing the detection of Merapi plume detection has been removed.

Overall, I think that Section 4 is not very clear and should be reorganized. It is noted that it starts with a long introduction from lines 248 to 314, followed by only one subsection (4.1). Section 4 has been reorganized as follows: 4 Volcanic plumes and quiescent

[Figure]

periods 4.1 Quiescent period 1997 - 2003 4.2 Volcanically-active period 2003-2013 4.2.1 Detection of Sarychev and Nabro plumes 4.3 Post-Nabro period 4.4 Identification of volcanically-perturbed periods The text in Section 4.4 on the method for selection of volcanic periods has been revised (see above).

Section 5: Line 443: please describe Fig 7c. Second sentence in the last paragraph of Sect. 5 has been modified: "Fig. 7c shows annual cycle of water vapour vertical profile, providing further evidence to this finding."

Please also note the supplement to this comment:
http://www.atmos-chem-phys-discuss.net/acp-2016-846/acp-2016-846-AC2-supplement.pdf

---

## Author Comment (AC3) · 4 Jan 2017

Please check the pdf file in supplement, which contains the formatted text for easier reading.

We thank Anonymous Referee #2 for providing useful remarks on the manuscript.

The English syntax has been re-checked by native English-speaking co-authors.

Title/Abstract: Since this article is submitted to the NDACC 25th anniversary Special Issue, it would be suitable to explicitly cite NDACC either in the title, or in the abstract (when presenting the OHP lidars). The first sentence of the abstract was modified: "...obtained by two independent regularly-maintained lidar systems operating within

the Network for Detection of Atmospheric Composition Change (NDACC)."

Section 2.3: Are all extinction coefficient conversions factors (ke) using the same reference of Jäger and Deshler (2002; 2003)? Please specify if so. The value of wavelength exponent depends on the wavelength pair. For the 355 nm - 532 nm pair ke value was adapted from Jager and Deshler (2002). The same value (ke=-1.6) was used for SAGE II (525 nm) and GOMOS (550 nm), for which the wavelength conversion means multiplying the extinction by a factor of 0.979 and 1.055 respectively. For OSIRIS (750 nm) and OMPS (675 nm) the wavelength exponents were suggested by the instrument PIs. Following a suggestion of Referee#1, a new table (Tab. 1) containing information on the wavelength exponents $\kappa$e used for conversion of all data set was added to Sect. 2.3.

Line 200 and Fig. 1: The number of CALIOP samples is much larger than the number of samples from the other instruments. Why not restricting the CALIOP coincidence window within the OHP region instead of the full zonal-mean? This would (maybe) improve the agreement with the lidars, for example during the periods 2007-2008 and 2010-2011. Measurements of stratospheric aerosol by CALIOP, particularly during background conditions have a low signal to noise ratio and require substantial averaging to be useful. On a monthly-mean scale the aerosol loading in the stratosphere, even under active volcanic conditions is rather uniform thanks to a strong zonal flow. This is confirmed by a strong correlation and high degree of agreement between all the sAOD series. Please note also that the OHP lidar data have been reprocessed over the course of paper revision (see Author comment in ACPD, AC1 "Reprocessing of the OHP lidar data and related changes to the manuscript"). This has improved the qualitative results of intercomparison between sAOD1730 from OHP lidars and CALIOP (see updated Tab. 2, former Tab.1). For example correlation between LiO3S and CALIOP has increased from 0.85 to 0.91. The mean difference between LTA and CALIOP is -0.4 $\pm$ 1.7% with correlation of 0.96. Such figures leave very limited room for further improvement of the agreement.

Fig. 1: Can the aspect ratio of this figure be less elongated (more square) in order to distinguish the various measurements from each other? The aspect ratio of Fig. 1 has been reduced.

Lines 285-287: Fig. 3 does not really show a difference in the e-folding rate for Sarychev and Nabro, it looks more like the background level after Nabro (late 2012) is simply higher than that after Sarychev (early 2010). The authors refer to time-series from CALIOP and OSIRIS . Can they refer to specific publications? While it may occur from Fig. 3 that the decay time of volcanic aerosol after Sarychev and Nabro eruptions is about the same, sAOD after Nabro decreases to 1/e of the peak sAOD value only after 19 months according to the zonally-averaged satellite series. Over the course of the year 2012, the remnants of Nabro aerosol are still residing in the stratosphere and this period is classified as volcanically-perturbed. The "true" background level is achieved only in early 2013. The duration of the volcanic period associated with Nabro eruption is determined using the criteria described in Sect. 4.4 (former Sect. 4.1). We could not find in the literature any concrete statements regarding the full lifetime of Nabro aerosol at mid-latitudes, however the observations provided here serve well to address this question.

Lines 329-330 and fig.3: How do the monthly-mean sAOD uncertainties compare with the 1-sigma threshold level set to determine what is quiescent and what is not? Although there is a risk to overload Fig. 3, it would be interesting to overplot vertical bars to denote +/- 1 standard uncertainty for each monthly mean sAOD sample plotted. In any case, some text should be added to discuss the relative magnitude of this uncertainty and the 1-sigma threshold value. This will determine whether the observed increase can be considered statistically significant or not. The average standard uncertainty for the monthly-mean values of sAOD1730 from OHP lidars amounts to 4.8% (LiO3S) and 3.5% (LTA). The 1-sigma threshold level for the "reference" quiescent periods computed on the base of monthly mean values is 12.6%. The sentence in the first paragraph of Sect. 3 was modified to explicitly mention the standard errors of monthlymean values from OHP lidars: "The average error for monthly averages of OHP lidars' sAOD1730 amounts to 4.8% (LiO3S) and 3.5% (LTA)." Following a similar remark from Referee #3, the sentence in the end of the second paragraph in Sect. 4.1 (former Sect. 4) was modified as follows: "According to the mean of OHP lidars, the average background sAOD1730 for the "reference" quiescent period of 2.37•10-3 ± 12.6% (1$\sigma$), which is marked in Fig. 3 by dashed line and grey shading, indicating ±1-$\sigma$ range of values. SAGE II reports sAOD1730 for the same period of 2.4•10-3 ± 10.2%."

Lines 440-441: See above comment on line 200: For CALIOP, it would make more sense to use an average over a longitude band centered over OHP (e.g., +/-20 deg) rather than a full zonal mean. After reprocessing of OHP lidar data (please see AC1 for details) it was possible to extend the lower boundary of all the plots showing aerosol vertical distribution below 15 km. The updated Fig. 7 shows better the signatures of ATAL, both in OHP and CALIOP panels, rendering the interpretation less ambiguous. Restricting CALIOP data to a limited longitude band would not necessarily enhance the ATAL signal. As stated in Sect. 3 (Intercomparison of OHP lidars and satellites sounders), the coherence between lidar and satellite series suggests that the stratospheric aerosol burden is zonally-uniform at least on a monthly-mean scale. This is explained by the presence of strong zonal winds in the stratosphere, which rapidly homogenize the aerosol and tracers in the zonal direction. OHP is under influence of Asian monsoon circulation during July-September season. In October, the mean zonal flow intensifies at the level of ATAL as can be seen in Fig. AR2.1, and stratospheric aerosol becomes zonally-uniform at OHP latitude.

Last Paragraph (Lines 558-563): The authors should also emphasize on the critical need for ground-based lidar observations in the next decade, as there will possibly be a gap in aerosol profilers from space after CALIOP has ceased operation. The last paragraph has been updated with the following sentence: "The need for continuous ground-based observations becomes critical as there may be a lack in space-borne aerosol measurements after CALIOP has ceased operation."

Please also note the supplement to this comment:
http://www.atmos-chem-phys-discuss.net/acp-2016-846/acp-2016-846-AC3-supplement.pdf

[Figure]

100 -125 hPa,  September 2012

**Fig. 1.** Figure AR2.1top. Wind field at 100-125 mBar level from ERA-Interim reanalysis during September 2012.

[Figure]

**Fig. 2.** Figure AR2.1bottom. Wind field at 100-125 mBar level from ERA-Interim reanalysis during October 2012.

---

## Author Comment (AC4) · 4 Jan 2017

Please check the pdf file in supplement, which contains the formatted text for easier reading.

We thank Anonymous Referee #3 for a focused attention to our study and a detailed review.

Before reading this reply please refer to the Authors' comment published in the interactive discussion: AC1 "Reprocessing of the OHP lidar data and related changes to the manuscript".

The revision applied to the data and the updated discussion thereof are expected to

help reducing the skepticism expressed by the Referee #3. However, in the reply that follows we tried to thoroughly address each remark, providing occasionally figures that support our interpretation. We also emphasize that our interpretation of mid-latitude aerosol variability relies on the existing and widely accepted studies.

265-267. The authors state, "Both SAGE II and OHP lidars report an average background sAOD1730 for the "reference" quiescent period of 2.3• 10-3 $\pm$ 2.4% (2 SE), which is marked in Fig. 3 by dashed line and grey shading, indicating 1-$\sigma$ range of values." This statement implies that sAOD should be between 0.00225 and 0.00235 since 2.4% of 0.0023 is 0.00005, yet the range shown in the figure is much larger than this. The authors claim that 2.4% is 2 SE, which, the reader is left to assume, means 2 standard errors. Then the authors say the shading represents 1-$\sigma$, without explanation. So in the end the reader is unsure what is shown in the figure, but it seems to be larger than 2.4% of the mean value quoted and how does 1-$\sigma$ compare with 2 SE? The sentence was supposed to mean that the average background sAOD1730 is 2.3•10-3 $\pm$ 2.4% (2 SE), where 2.4% is two times relative standard error obtained as standard deviation of monthly-mean values of sAOD1730 during the background period divided by a square root of the number of these values and expressed in percentage from the average background sAOD1730 (2.3•10-3). The standard deviation itself amount to 12.6% (OHP lidars) and 10.2% (SAGE II). To avoid confusion the mention of standard error was replaced by standard deviation and the sentence was modified as follows: "According to the mean of OHP lidars, the average background sAOD1730 for the "reference" quiescent period of 2.37•10-3 $\pm$ 12.6% (1$\sigma$), which is marked in Fig. 3 by dashed line and grey shading, indicating $\pm$1-$\sigma$ range of values. SAGE II reports sAOD1730 for the same period of 2.4•10-3 $\pm$ 10.2%."

231-234. It is very difficult for the reader to understand how the figure supports these statements. Above 25 km the lidar data do not show any particularly different bias compared to satellite than below in the left panel of Fig. 2. The lidar data lie within the symbols for both SAGE and OSIRIS. On the right panel the lidar data split the

satellite data and the agreement is overall better than below 25 km. Below 25 km the agreement with CALIOP remains good but is worse OSIRIS and OMPS. After the reprocessing of OHP data (See AC1 "Reprocessing of the OHP lidar data and related changes to the manuscript"), Fig. 2 and associated description have been fully revised: "Figure 2 displays a comparison of aerosol extinction profiles averaged over two 20-month volcanically-quiescent periods 2002-2003 and 2013-2014 covered by time-overlapping observations by two different triplets of satellite sounders. The comparison reveals close agreement between OHP lidar, SAGE II, GOMOS and OSIRIS (Fig. 2a) above 15 km and somewhat poorer agreement below. Fig. 2b suggests a good agreement between OHP lidar and CALIOP (relative difference 5-10%) throughout the entire range of altitudes except the uppermost layer above 25 km, where OHP lidar is 15-20 % low with respect to CALIOP. This feature may be related to an error in lidar calibration, relying on the assumption of the absence of aerosol above 30 km, which – as suggested by CALIOP data calibrated at higher altitudes - may not always be the case. The other two satellite sounders covering 2013-2014 period – OSIRIS and OMPS - show somewhat larger discrepancies (reaching 30% ) with OHP lidar and CALIOP in the uppermost and lowermost layers. This discrepancy may be due to the use of a fixed lidar ratio and wavelength exponents, which may vary with height depending on the size distribution of aerosol."

288-301. Surely the differences between the plumes of Sarychev and Nabro are primarily driven by the significantly different latitudes of the two eruptions, compared to the latitude of OHP, and the dominance of the mixing by zonal flow in the stratosphere. Sarychev, at nearly the same latitude as OHP, is detected very early and the volcanic plume appears as pulses of aerosol, as these pulses are advected around the Earth before they are significantly mixed by the general flow. In contrast the aerosol from Nabro is already well mixed by the general flow prior to its arrival at OHP, 45 days after the eruption. To effectively compare the evolution of these two eruptions the color scales should be adjusted to both start at the day of first detection of Nabro, 45 days after each eruption. All profiles prior to this time from Sarychev could be indicated as

black profiles.

After a careful profile-by-profile data screening (see AC1), the considerations on the detection timing of Sarychev and Nabro plumes have been entirely revised. As a matter of fact, the delay between eruption and plume detection at OHP is the same for Sarychev and Nabro eruptions. The first unambiguous detection of Sarychev plume (erupted 12.06.2009) dated 26.06.2009, whereas Nabro plume (erupted 12.06.2011) was detected on 28.06.2011, that is 15 days after the eruption (cf. 45 days reported previously). Figure 4 was updated with the profiles previously discarded and the colour scale was uniformed for both Fig.4a and 4b. The respective description in Sect. 4.2.1 (former Sect. 4) has been revised as follows: "A better insight into the temporal evolution and vertical structure of Sarychev and Nabro plumes is provided by Fig. 4, showing scattering ratio (SR) profiles obtained by OHP LiO3S lidar during the corresponding volcanic periods and converted to 532 nm. The plume of Sarychev was detected at OHP 14 days after the eruption as sharp SR enhancements in the lowermost stratosphere reaching a maximum value of 4.8 at 15 km (30.06.2009). On 15.07.2009 a sharp enhancement in SR of 2 was observed by LiO3S as high as 21.7 km. The presence of aerosol at this level is confirmed by LTA observations on the next night (not shown), which reported SR at this level reaching a value of 3.5. A remarkable scatter between the individual profiles points to a rapid three-dimensional evolution of the plume (Jegou et al., 2013), dispersed by the stratospheric mean zonal flow, which reversed over the course of the plume permanence. The first signatures of Nabro plume were detected at OHP already 15 days after the eruption: a strong peak in SR reaching 2.8 was observed at 16.5 km on 28.06.2011 (Sawamura et al., 2012). Over the course of July, several relatively thin (<1 km) aerosol layers with SR below 1.6 were detected between 14 and 17 km altitude. Starting from early August (∼50-60 days after eruption) the plume of Nabro – as observed at OHP – expands in altitude and obtains a smoother shape indicating the arrival of air masses, in which the aerosol-laden air is mixed with the ambient air by the general flow. Broad (∼3 km) enhancements in SR of ∼1.5 centered at 17 km were observed at OHP through March 2012."

319-320. "The plumes of more distant (tropical) eruptions are not always obvious in sAOD series." What is a more distant tropical eruption? Nabro is tropical. Considering the dominant zonal flow does the longitude of a tropical eruption make a big difference? Why are these "more distant tropical eruptions" not evident in sAOD series? Is this sAOD now meant to only imply sAOD at OHP? Distant and close tropical eruptions will make a difference in sAOD depending on where sAOD is measured, but the reader is left to guess what is intended. The text implies that the plume from a volcanic eruption has a rather direct stratospheric transport to the mid latitudes from a tropical eruption, but doesn't the dominant zonal flow in the extra tropical stratosphere confound this idea? Technically, among all the VEI=4 eruptions since 1994 Nabro is closest to OHP in absolute distance. Also, as can be inferred from the analysis of dispersion of Nabro plume (e.g. Sawamura et al., 2012; Fairlie et al., 2014) the longitude of eruption may make a large difference, especially in the context of detection of volcanic aerosol at a mid-latitude site. Eventually, the dominant zonal flow would uniformly distribute the aerosol load longitudinally, however the efficiency of meridional transport of a volcanic plume depends strongly on the season and location of the eruption. A rapid transport of Nabro plume to Mediterranean region was ensured by the circulation around Asian monsoon. In contrast to that, aerosol from eruptions occurring elsewhere within the tropical belt would tend to remain mainly in the tropics, while their transport to Northern mid-latitudes would be inhibited during Boreal summer, when the subtropical mixing barrier is stronger. For this reason there may be a substantial delay between the eruption and the time when the aerosol-laden air has reached mid-latitudes. By that time, the air is already very well mixed with the environment and hence the associated enhancement in aerosol loading as observed at OHP appears "less obvious". This is what we refer to as an "aged" plume. The respective sentence was modified for clarity: "The plumes of more distant (tropical) eruptions are not always obvious in sAOD series OHP observations".

336-337 and Fig. 5. "Aged" is not a very descriptive term. Better would be some consistency such that the volcanic curves represent an average of the measurements

over some specified time period, which ideally would be the same time after each eruption. The term "aged" describes the time lag between an eruption and its detection at the measurement site. As discussed in response to the previous remark, the timing and location of an eruption plays a crucial role in how soon the volcanic aerosol is transported to OHP latitude. The altitude of volcanic injection is also important in this context. Aerosol from high-altitude injections (e.g. Soufriere Hills, Kelud), would remain in the stratosphere for a considerable period, however, the timescale of poleward transport of their plumes (mainly through Brewer-Dobson circulation) may be long. In contrast to that, the aerosol from volcanic injections into the TTL (e.g. Tavurvur) will be removed more rapidly (through mixing and/or cloud scavenging), but its transport to mid-latitudes may be faster thanks to more efficient meridional exchange at those levels. Thus, each eruption requires an individual approach, which makes use of both global and local observations.

353-357. The CALIOP data are far from clearly supporting the suggestion that the plume from Merapi was observed at OHP. The structure in the CALIOP data at OHP latitude in early 2011 which coincides with the blue shading in the OHP data has an origin prior to Merapi, whereas it is not obvious that the plume from Merapi is still intact at 45ÅŮeN. The sAOD1519 from CALIOP is 2e-3 to 3e-3 compared to 5e-3 at OHP. In contrast after Nabro in mid to late 2011 the CALIOP data display a significant increase in aerosol at OHP latitudes whereas OHP sAOD is hardly larger than the value attributed to Merapi. Such discrepancies raise questions about how well these two data sets really agree, particularly at these altitudes. Is this reflective of the differences between the OHP and CALIOP measurements below 16 km in Fig. 2b. This seems unlikely. Figure 6, in the discrepancies of the timing between OHP sAOD and CALIOP sAOD for both Nabro and Sarychev, raises question about the correspondence of these two data sets. At the very least the timing of Sarychev, Nabro, and many of the aerosol minima appear to be displaced, with OHP lidars lagging the CALIOP data. Indeed, it is not obvious that Merapi plume has actually reached the OHP latitude. CALIOP data show the northern boundary of the plume at around 40° N. OSIRIS data suggest

that Merapi plume did not extend beyond 35° N. The reason why Merapi plume was said to be observed at OHP was that all criteria for selection of volcanically perturbed periods were fulfilled for this eruption, namely a) propagation of plume to the Northern mid-latitudes (as inferred from CALIOP) and b) criteria i) and ii) regarding sAOD and SR (Sect. 4.4, former Sect. 4.1). However, after reprocessing of OHP data (see AC1) the criteria applied to the lidar data are no longer fulfilled. Indeed, the reprocessed lidar series do not provide indication for the increase of aerosol load around the turn of 2011 and this period is no longer considered as volcanically-perturbed. As far as the consistency between OHP and CALIOP series is concerned, the sAOD1519 series from OHP lidar and CALIOP are in fact in good agreement. The upper panel of Fig. AR3.0 shows the CALIOP curve, revealing itself in good correlation with OHP series. We note that during the periods of high aerosol load (posterior to Ok/Ka, Sa and Na eruptions) CALIOP shows smaller AOD1519 values compared to OHP lidar. In CALIOP retrieval the attenuation due to aerosol is not corrected for. However, with the two-way transmission near 0.97-0.98 (for AOD 0.010-0.015) the error on the AOD at 15 km would not be bigger than 2-3%.

Figure AR3.0. Time series of monthly-mean sAOD1519 from OHP LiO3S lidar and CALIOP (top) and time-latitude section of sAOD1519 from CALIOP in log-scaled color map with indications of VEI 4 eruptions (bottom). Time periods considered as perturbed by volcanism (Tab. 2) are shaded light blue in the top panel. White arrows (in 2007-2008) represent the mean meridional component of monthly/zonally-averaged horizontal wind at 100 hPa from ERA-Interim reanalysis. Dashed and dotted contours depict zonal-mean water vapour mixing ratio at 100 hPa from Aura MLS.

367-372. Fig. 6 displays 10 years of CALIOP AOD from 15-19 km from 60S to 60 N. What fraction of the troposphere is included here? Certainly in the equatorial and tropical regions there is about 1-2 km of tropospheric data since the tropopause is typically near 17 km. The upper troposphere can be quite clean if there is deep convection or it can be influenced by tropospheric aerosol. To attribute all the data shown in Fig. 6 to

the stratosphere is misleading. Here the authors want to suggest based on signatures, clouded by the uncertainties just mentioned, that 4 of these 10 years display evidence of the ATAL. But how would the ATAL be separated from other aerosol laden air from the upper troposphere? What other evidence is there to link this slight change in AOD to the ATAL? Is it really so clear in terms of the timing of these events? How similar is it? Finally this is a paper about the OHP lidar record not a broad scale interpretation of the CALIOP data from 60 S to 60 N. If the latter is the intent then do a complete job on the CALIOP observations. Here the intent appears to be on the OHP lidars. If so then there should be a better discussion of when the CALIOP is in agreement with OHP, when it is not, and why there are differences.

We do not mean to attribute all the data in Fig. 6 to the stratosphere. In the tropics, 15-19 km layer includes a part of the TTL, although mostly above LZRH – Level of Zero Radiative Heating – above which the air tends to rise (Fueglistaler et al., 2009). Meanwhile, at OHP latitude it is entirely in the lower stratosphere. The figure is intended to show the processes that affect the variability of stratospheric aerosol at mid-latitudes, whatever layer of the atmosphere is at play. Notation "sAOD1519" for CALIOP in Fig. 6 is chosen for compatibility with that for OHP. For the sake of better precision it was changed to AOD1519 for CALIOP. ATAL signatures in CALIOP data in Figure 6 alone would not allow us to unambiguously attribute the slight positive anomalies in aerosol to ATAL. However, if considered together with the available knowledge of ATAL's three-dimensional extent and seasonality available from the literature (e.g. Vernier et al, 2015) and commonly accepted, this feature can hardly be attributed to anything else but the phenomenon of ATAL. As an additional support to our statement Fig. AR3.1 shows CALIOP AOD for the layer 15-16.5 km, where ATAL occurrence is more prominent. We do not intend to provide a broad scale interpretation of CALIOP data from 60S to 60N; this can be found elsewhere in the literature quoted throughout our paper. The rationale behind showing the global distribution of AOD is threefold: i) to show that background aerosol at mid-latitude is modulated by the global circulation, particularly poleward transport; ii) to identify the processes responsible for the annual cycle

of aerosol at mid-latitudes, laying the ground for interpretation of Fig. 7; iii) to point out the similarity in aerosol and water vapour time-latitude patterns, suggesting the same driver for the both.

Figure AR3.1. Time-latitude section of AOD in 15-16.5 km layer from CALIOP in log-scaled color map with indications of VEI 4 eruptions. Systematic increase of AOD in the Northern sub-tropics/mid-latitude is attributed to ATAL.

373-382. This picture is a bit less clear than suggested. Many of the Northern Hemisphere low aerosol tongues are rather discontinuous even when volcanoes are not involved. The lidar and CALIOP timing of the low aerosol load are different. While there is some evidence for the author's assertion, it is far from definitive, and other processes may be involved. The influence of the troposphere on the AOD displayed is unclear. It is also not clear to what extent a higher summer tropopause would affect the OHP data compared to a lower tropopause in the winter. If the authors wish to pursue this type of interpretation of the CALIOP data they should consider preparing a paper focused on such analysis of the CALIOP data and not add it as a sidelight to this paper about OHP lidars. Measurements of very low aerosol concentration are prone to large error, even when zonal/monthly means of CALIOP are used. This is why the tongues may sometimes appear discontinuous. In order to support the interpretation of clean air tongues, we overplot the water vapour pattern, which emphasises the poleward transport of dry (clean) air, whose composition is set during austral summer. There is no discovery here, we rely on the previous studies and make sure to properly refer to them. We do not mean to provide a breaking explanation of the aerosol minimum during Austral summer in the tropics, this is already done by Vernier et al., (2011) and widely accepted (Kremser et al., 2016). Alternative processes that might potentially responsible for the LMS aerosol minimum at mid-latitudes are discussed in Sect. 6.

Fig. 7a and 7b display several discrepancies. CALIOP data display the expected Junge layer with minimums below 18 and above 24 km, and a maximum near 20 km throughout the year. OHP suggests a significant modulation of the Junge layer with

a decrease of AOD from 1.08 to 1.04 from April to December which is not seen in the CALIOP data. Is this seen in other data sets? It is not clear what would cause this modulation of the Junge layer. The CALIOP data do not show a strong increase in aerosol near 16 km in the autumn. The authors explain this away as due to zonal averaging. But really is the connection so immediate, from the Asian monsoon to 45ÅŮęN, that the ATAL would only appear in the OHP data? Is the ATAL signal so small that it is diluted with the zonal average, even though that average would incorporate much more of the Asian monsoon outflow than would reach OHP?

After the reprocessing of OHP lidar data (see AC1) it was possible to extend the analysis of aerosol profiles down to the tropopause. Fig. 7 has undergone a major revision, namely: i) altitude range extended down to 13.5 km in all panels; ii) color scale range in Fig. 7a,b reduced to SR=1.1 for emphasising the background aerosol pattern; iii) CALIOP data in Fig.7b restricted to 45°±2.5° N (see Fig. AR3.2). In addition, slight change to the aerosol pattern in Fig. 7a,b is due to revision of the volcanic mask (removal of Merapi, see above). In the updated Fig. 7a,b the ATAL signature is better pronounced both in OHP and CALIOP data. It is noted that the onset of ATAL layer occurs earlier in the CALIOP section, which might just be due to zonal averaging, which includes the north-east part of ATAL.

Figure AR3.2. Climatological month-altitude sections of a) SR from OHP LiO3S lidar for volcanically-quiescent periods over the entire measurement time span (1994-2015); b) zonal-mean SR at 45° N ±2.5° from CALIOP, June 2006 - September 2015 for volcanically-quiescent periods (Tab. 2) The discussion around Fig. 7a,b has been revised as follows: "Fig 7b provides a satellite zonal-mean view on the non-volcanic aerosol annual cycle observed by CALIOP since 2006. The month-altitude pattern of zonal-mean background aerosol revealed by CALIOP supports the climatology observed by OHP lidar. The main features, namely the winter maximum of the Junge layer upper boundary, the spring maximum of SR in the middle layer (19-25 km) and the upward propagation of the late-spring clean feature are readily discernible in both

OHP and CALIOP climatologies. The signature of ATAL at 15-16 km altitude is also well pronounced in CALIOP section, which shows its maximum development in August as opposed to September according to OHP climatology. This may be due to zonal averaging for CALIOP, which incorporates the mid-Asian part of Asian monsoon, where ATAL is better developed in August (Fig. 2 in Vernier et al., 2015). OHP lidar and CALIOP capture well and agree on the main features of background aerosol annual cycle in the lower mid-stratosphere, whereas above 25 km CALIOP shows higher SR values compared to OHP lidar and somewhat less pronounced annual cycle. This may be due to higher altitude of calibration for CALIOP retrieval and the use of different atmospheric models for deriving molecular backscatter (Sect. 2.3 and 3). "

Why are the time periods covered by Fig. 7a, 7b so different? Is there a point to be made about similarities of any non-volcanic period, or is the point to show how similar the OHP lidars are to CALIOP? If the latter then wouldn't it be better to compare the same time frames? We use the full time span of OHP data for constructing the annual cycle of background aerosol in Fig.7a to enhance the sampling and to reduce the noise. The pattern remains essentially the same if we consider the same time periods for OHP and CALIOP (see Fig. AR3.3). There is a respective mention in Sect. 5.1, end of paragraph 3: "Importantly, for any quiescent subperiod over the course of 22 yr OHP series, the pattern is essentially the same." The more important point of comparison between OHP and CALIOP annual patterns is to show how similar they are.

Figure AR3.3. Climatological month-altitude sections of SR from OHP LiO3S lidar for volcanically-quiescent periods over the CALIOP observations period (2006-2015).

525-526. Calling the authors' explanations for the observations "rather robust" is not justified in this reviewer's mind, and suggesting there may be alternate explanations, which are not explored, but should be, is less than genuine at this point in the conclusions. The phrase "rather robust" refers to the main point that is made in the paragraph regarding the control of mid-latitude background aerosol by convective processes followed by poleward transport of clean and polluted air, which represents the main driver of aerosol annual cycle at OHP latitude. We have provided a sufficient amount of observational evidence to this finding using both global and local measurements after having demonstrated the consistency between the both. When referring to the convective processes responsible for cross-tropopause transport of clean or polluted air, we rely on the existing and widely accepted studies. Further, we do consider and discuss the alternative contributors to the observed aerosol annual cycle in the two paragraphs that follow. The last sentence in paragraph 3, Sect. 6 has been modified: "Although this interpretation appears self-consistent, alternative contributors should also be considered."

The discussion section is a recap of the conclusions reached based on the analysis discussed above which I find incomplete and perhaps misleading. The models the authors have to characterize the data are too simplistic and ignore many complicating factors. Section 6 (Discussion and summary) complies with its purpose: to discuss alternative interpretations and to provide a recap of conclusions.

870. embedded panel? Do the authors mean the legend? There is in fact no embedded panel in Fig. 1. The mention of it was removed from the figure caption.

175. I am not quite sure what is meant by occultations for a limb scatter instrument. What is being occluded? The word "occultations" was replaced by "vertical profiles".

291. 3.4 units? Do the authors mean a scattering ratio of 3.4? Yes, we meant "scattering ratio of 3.4". The text was corrected accordingly.

307-308. Why do the satellite measurements not agree with the optical depth decrease after January 2015 observed by the OHP lidars? Rather the satellites remain elevated at the January level. A minor and transient decrease of optical depth after January 2015 seen by OHP lidars (Fig.3) is also resolved by the satellite mean series but appears somewhat less pronounced. Note that this decrease in also less pronounced in the revised OHP series.

[Figure]

309. This comment on Calbuco is not really necessary here since it does not affect post Nabro OHP and forces the reader to look ahead to Fig. 6 to verify the statement, which is then called out of order. The mention of Calbuco eruption has been removed from the paper.

323. What is the partial sAOD examined? Is it the same for all satellites? It should be stated what the AOD covers. The sentence has been modified: "The plumes were detected by examining time-latitude sections of sAOD1730 and AOD1519 from all satellite records..."

324. Another call out to Fig. 6 out of order. Should the figure orders be reversed? The introduction of Fig. 6 in this section is indeed premature, however it provides a great aid in understanding how the volcanic plumes are detected using satellite data. The sentence has been nevertheless modified : "...(example for CALIOP is provided hereinafter in Sect. 5). The figure order can not be reversed as this would strongly disrupt the flow of presentation.

329-332. "monthly-mean sAOD1730 and SR" where? Is this for OHP only or does it include all the satellite data? In ii) specify the "reference" quiescent period, e.g. 1997-2003. This is for OHP only. The preceding sentence has been modified: "In this way, a period is considered as volcanically-perturbed if both of the following two conditions are fulfilled in OHP observations:..". Reference period has been specified in ii): "ii) monthly-mean SR profile exceeds the 1-$\sigma$ range of the "background" SR profile - an average over the entire "reference" quiescent period of 1997-2003..."

336. Concerning the quiescent period, the text and Fig. 5 caption state 1997-2003, the legend in the figure states 1998-2003? These should be consistent. The legend in Fig. 5 has been corrected to 1997-2003.

365-366. "The enhanced poleward transport into the winter hemisphere is exhibited by meridional wind vectors in Fig. 6." Then according to the figure there is no meridional wind after 2009. Is this correct? No, this is not correct. The figure caption says:

"White arrows (in 2007-2008) represent the mean meridional component of...". The absence of wind vectors after 2009 does not mean the absence of meridional wind in the atmosphere. The plotting of wind vectors was period-limited to avoid overloading the figure.

Fig. 7 caption. The reader does not know what is meant by "SR from OHP LiO3S lidar for selected volcanically-quiescent periods . . ." What is the selection based on? Is it all non-volcanic periods or just select periods? By "selected" we meant that the periods were selected on the base of criteria described in Sect. 4.4 (former Sect. 4.1). The word "selected" was removed from Sect. 5.1 and Fig. 7 caption.

428-429."Importantly, for any quiescent subperiod over the course of 22 yr OHP series, the pattern is essentially the same." The OHP lidar record is only 22 years long, so what does this statement mean? Do the authors mean any quiescent subperiod within the 22 year data record? Yes, we mean "any quiescent subperiod within the 22 year data record". The sentence has been modified: "Importantly, for any quiescent subperiod within the 22 yr OHP record, the pattern is essentially the same."

Please also note the supplement to this comment:
http://www.atmos-chem-phys-discuss.net/acp-2016-846/acp-2016-846-AC4-supplement.pdf

[Figure]

[Figure]

**Fig. 1.** Figure AR3.0

CALIOP AOD 15-16.5 km zonal mean

Fig. 2. Figure AR3.1

**OHP LiO3S, SR@532 nm, 1994-2015 quiescent**

**Fig. 3.** Figure AR3.2top

**CALIOP SR@532 nm, 45º N ± 2.5º, 2006-2015 quiescent**

**Fig. 4.** Figure AR3.2bottom

**OHP LiO3S, SR@532 nm, 2006-2015 quiescent**

**Fig. 5.** Figure AR3.3

---

## Author Comment (AC5) · 4 Jan 2017

We thank Dr. M. Fromm for bringing up the issue in Fig. 4a showing the detection of aerosol plume from Sarychev eruption. This remark led us to carefully revisit the OHP lidar data series. As a matter of fact the entire data set was subjected to reprocessing (see AC1 "Reprocessing of the OHP lidar data and related changes to the manuscript"). It was found in particular that a semi-automated screening procedure applied to the initial version of the data leaves behind some of the useful measurements, e.g. a strong aerosol peak at 21.5 km originating from the Sarychev eruption or an early detection of Nabro plume (15 days after the eruption as opposed to 45 days reported initially). Figure 4 and the discussion around it (Sect. 4.2.1) were completely revised.

[Figure]

Figure 4. Individual (coloured curves) and period-averaged (black circles) scattering ratio profiles from OHP LiO3S lidar acquired after the eruptions of Sarychev (a) and Nabro (b) volcanoes. The colours of individual profiles denote the days since eruption. The eruption dates and plume detection periods are indicated in each panel. Only the data above the local tropopause (NCEP) are shown.

Please also note the supplement to this comment:
http://www.atmos-chem-phys-discuss.net/acp-2016-846/acp-2016-846-AC5-supplement.pdf

[Figure]

**Fig. 1.** Figure 4a

[Figure]

Nabro 13° N (13.06.2011)
Detection at OHP: 28.06.2011 - 18.03.2013

**Fig. 2.** Figure 4b